# Synthetic History: Evaluating Visual Representations of the Past in Diffusion Models

**Maria-Teresa De Rosa Palmini**
University of Zurich
Zurich, Switzerland
`maria.teresa.derosapalmini@uzh.ch`

**Eva Cetinić**
University of Zurich
Zurich, Switzerland
`eva.cetinic@uzh.ch`

## Abstract

As Text-to-Image (TTI) diffusion models become increasingly influential in content creation, growing attention is being directed toward their societal and cultural implications. While prior research has primarily examined demographic and cultural biases, the ability of these models to accurately represent historical contexts remains largely underexplored. To address this gap, we introduce a benchmark for evaluating how TTI models depict historical contexts. The benchmark combines HistVis, a dataset of 30,000 synthetic images generated by three state-of-the-art diffusion models from carefully designed prompts covering universal human activities across multiple historical periods, with a reproducible evaluation protocol. We evaluate generated imagery across three key aspects: (1) Implicit Stylistic Associations: examining default visual styles associated with specific eras; (2) Historical Consistency: identifying anachronisms such as modern artifacts in pre-modern contexts; and (3) Demographic Representation: comparing generated racial and gender distributions against *LLM-estimated historically plausible demographics*. Our findings reveal systematic inaccuracies in historically themed generated imagery, as TTI models frequently stereotype past eras by incorporating unstated stylistic cues, introduce anachronisms, and fail to reflect plausible demographic patterns. By providing a reproducible benchmark for historical representation in generated imagery, this work provides an initial step toward building more historically accurate TTI models.

## 1 Introduction

Text-to-image (TTI) models have become powerful tools for generating high-quality images from text, with applications in different domains such as art, education, and media (Vartiainen & Tedre, 2023; Maharana et al., 2022). However, recent critical analyses of TTI models highlight various issues, most notably the reinforcement of demographic biases, particularly gender and racial stereotypes (Chauhan et al., 2024; Luccioni et al., 2024; Zhang et al., 2024b; Bianchi et al., 2023). In addition, studies increasingly focus on cultural biases, such as Western-centric defaults and limited global diversity in the depiction of objects and traditions (Ventura et al., 2023; Senthilkumar et al., 2024).

While these studies primarily focus on present-day representations, how TTI models depict the past remains a largely underexplored area. Historical representation is not merely a matter of factual accuracy; it also reflects cultural memory, collective identity, and understandings of societal change. Synthetic imagery presents well-documented risks, such as the amplification of biases and stereotypes, the overlooking marginalized narratives (Sarhan & Hegelich, 2023), or the spread of misinformation (Dufour et al., 2024). When these issues appear in historically themed generated content, they can distort public understanding of the past and undermine the integrity of cultural memory. Despite these risks, current evaluations often overlook how such distortions emerge in visual depictions of history.

Existing approaches to evaluating historical accuracy in foundation models mainly focus on identifiable landmarks (Weyand et al., 2020), prominent historical figures (Lee et al., 2023), or objective description of archival imagery (Akbulut et al., 2025). However, such approaches overlook a more fundamental question: how do TTI models construct visual interpretations of the past? The widely publicized case of Google's Gemini (Team et al., 2024), which generated racially diverse Nazi

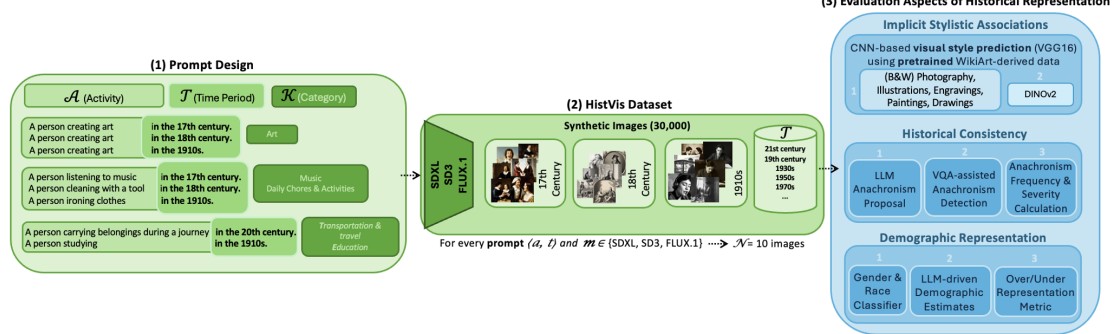

Figure 1: Overview of the benchmark: (1) Prompt Design, (2) HistVis Dataset of synthetic images, and (3) Evaluation of stylistic bias, historical consistency, and demographic representation.

soldiers, referred to as the "Black Nazi Problem" (Jacobi & Sag, 2024), highlighted the challenges of maintaining historical accuracy in AI-generated imagery and sparked debates on bias mitigation. Although AI bias is often discussed in terms of human representation, it also extends to the visual characteristics of generated images, such as the tendency of TTI models to associate certain historical periods with specific styles. Offert (2023), for instance, describes the difficulty of generating color images of 1930s fascist parades without outputs defaulting to visual features of early Kodachrome or colorized black-and-white photography. Besides demographic and stylistic biases, TTI models often fail to preserve chronological coherence, resulting in anachronistic depictions of objects, settings, or behaviors, such as smartphones in 18th-century scenes. Evaluating the historical reasoning of TTI models therefore requires a multifaceted approach that considers not only factual accuracy, but also implicit demographic assumptions, stylistic defaults, and consistent chronological representations.

To address these challenges, we introduce HistVis, a dataset of 30,000 synthetic images generated from 100 curated prompts covering 20 universal human activities (e.g., art, music) across five centuries (17th–21st) and five decades of the 20th century. The dataset is produced using three open-source diffusion models: Stable Diffusion XL (SDXL) (Podell et al., 2023), Stable Diffusion 3 (SD3) (Esser et al., 2024), and FLUX.1 Schnell (Black Forest Labs, 2025). Building on this dataset, we construct a benchmark that pairs HistVis with a reproducible evaluation protocol to assess how TTI models depict historical contexts along three dimensions: (1) Implicit Stylistic Associations, examining how models default to specific visual styles across time periods; (2) Historical Consistency, identifying anachronisms such as modern artifacts in pre-modern contexts; and (3) Demographic Representation, assessing whether portrayals of race and gender align with historically plausible patterns. Although our experiments focus on HistVis, the evaluation framework is model-agnostic and directly applicable to any other TTI model.

To the best of our knowledge, this is the first systematic evaluation of diffusion models in historical representation, offering a foundation for assessing representational fidelity and uncovering the underlying biases that shape how the past is portrayed in synthetic imagery.

## 2 RELATED WORK

TTI models have rapidly evolved from technical innovations into widely adopted tools with broad societal impact. As Cetinic (2022) argues, these models function as "cultural snapshots", encoding not only the culturally specific associations present in their training data but also the culturally contingent decisions made during model development. Consequently, they should be studied not just as tools but as artifacts that reveal tensions between neutrality, memorization and the politics of representation.

One significant dimension through which these tensions become evident is in demographic bias, particularly along gender and racial lines, which remains among the most extensively studied aspects of cultural encoding in TTI models. Studies consistently show that these biases manifest as stereotypical associations, such as overrepresenting women in caregiving roles and men in technical professions, thereby reinforcing existing social hierarchies (Seshadri et al., 2023; Hall et al., 2023b). Methodologically, these patterns have been identified through comparisons with real-world benchmarks such as labor statistics (Luccioni et al., 2024; Bianchi et al., 2023; Luo et al., 2024), embedding-based

bias metrics (Mandal et al., 2023), and classifier-based audits using models like BLIP-2 to evaluate responses to neutral occupational prompts (Cho et al., 2023). Recent work reveals that these biases not only affect the portrayal of gender but also associated objects and image layouts (Wu et al., 2024).

Beyond demographic biases, TTI models struggle with cultural representations, often reinforcing dominant narratives and missing regional nuance. Benchmarks such as CUBE Senthilkumar et al. (2024), UCOGC Zhang et al. (2024a), CulturalFrames (Nayak et al., 2025), and cross-cultural annotation studies (Hall et al., 2024) show that even culturally specific prompts (e.g., India's *gopuram*) frequently yield inaccurate depictions. Minor prompt changes, such as replacing a Latin character with a Greek or Arabic homoglyph, can shift outputs toward stereotypes (Struppek et al., 2023), while neutral prompts often produce Western-centric imagery with limited regional authenticity (Ventura et al., 2023; Hall et al., 2023a). Also, omitting country names can skew outputs toward U.S.-centric imagery, highlighting the need for globally aware, context-sensitive TTI models (Basu et al., 2023).

While demographic and cultural biases in TTI models have received considerable attention, historical representation remains a relatively underexplored dimension. In one of the few works to address the concept of history in TTI models, Offert (2023) discusses their stylistic biases and their tendency to produce visually reductive portrayals of the past. Related studies on AI-generated "promptography" show that models may produce convincing yet erroneous outputs that mimic historical artifacts, thereby revealing the constructed nature of these depictions (Martín Prada, 2025). Beyond image generation, hallucinated narratives in conversational or museum-based AI systems further shape public perceptions, amplifying the risk of historical distortion (Wills-Eve, 2023). Taken together, these findings indicate that while generative AI is capable of reconstructing historical imagery, it frequently does so in a selective, imprecise, and oversimplified manner.

Although some recent work started to address history-related tasks, these efforts tend to focus on recognition and factual alignment. For example, CENTURY (Akbulut et al., 2025) investigates how accurately and objectively multimodal models describe sensitive historical photographs, while HEIM (Lee et al., 2023) and the Google Landmarks Dataset v2 (Weyand et al., 2020) evaluate recognition of historical figures and landmarks. However, these existing benchmarks focus on factual correctness and identification rather than probing how models contextualize representations of the past.

Despite growing recognition of the socio-cultural implications of TTI systems, their historical grounding remains largely underexplored. Our work introduces a benchmark for evaluating historical representation in generated imagery, positioning historical depiction as a key dimension of cultural reasoning in generative models.

## 3   THE HISTVIS DATASET

As part of our benchmark, we introduce *HistVis*, a dataset of 30,000 synthetic images generated from prompts describing historically situated but universal human activities. Each prompt follows the neutral template *"A person [activity] in the [time period]"*, combining 100 activities $\mathcal{A} = \{a_1, \ldots, a_{100}\}$ with 10 historical periods $\mathcal{T} = \{t_1, \ldots, t_{10}\}$. Activities are grouped into 20 categories $\mathcal{K} = \{k_1, \ldots, k_{20}\}$ spanning domains such as art, religion, agriculture, work, and daily chores (Appendix B). We deliberately select activities that are timeless (e.g., farming, traveling) rather than tied to particular eras, so that cross-period variation reflects models' internal historical representations rather than prompt wording. By avoiding explicit references to figures, events, or objects, the design remains broadly applicable across cultures and minimizes the risk of encoding external biases.

Each activity is paired with 10 historical time periods $\mathcal{T} = \{t_1, \ldots, t_{10}\}$, spanning five centuries (17th–21st) and five decades of the 20th century (1910, 1930, 1950, 1970, 1990). For example, the prompt "A person listening to music" can be adapted to "in the 17th century" or "in the 1910s," enabling comparisons of how the same activity is rendered across eras. This selection balances broad historical coverage with finer temporal resolution: centuries capture long-term cultural trends, while 20th-century decades reveal more nuanced societal shifts. Although these choices serve our goals, alternative periodizations could highlight different aspects of historical representation. We use three state-of-the-art open-source diffusion models, SDXL, SD3, and FLUX.1 Schnell, chosen because they allow comparisons across both distinct architectures and successive versions of the same system. For each model $m \in \{\text{SDXL}, \text{SD3}, \text{FLUX.1}\}$, we generate $N = 10$ outputs for every activity–period pair $\langle a, t \rangle$, yielding a total of 30,000 synthetic images.

# 4 IMPLICIT STYLISTIC ASSOCIATIONS

The first aspect of our benchmark examines whether TTI models default to stylistic cues for historical periods, even when such features are not specified in the prompt. Although these implicit stylistic associations do not introduce factual errors, they can reinforce stereotypes and fixed assumptions about how different historical periods are "supposed" to look. Leveraging, therefore, the structure of the HistVis dataset where the activity remains fixed and only the time period varies, we introduce a methodology to quantify the strength and consistency of these associations across models and periods, enabling systematic analysis of a phenomenon previously described only qualitatively (Offert, 2023).

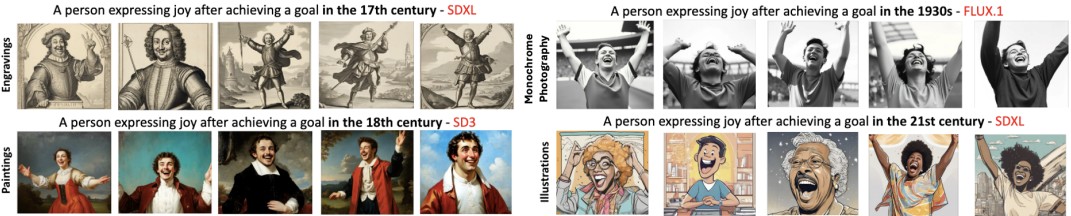

Figure 2: Examples of generated images reflecting different stylistic biases when a specific time period is added to the prompt "A person expressing joy after achieving a goal".

## 4.1 EXPERIMENTAL SETUP

To quantify implicit stylistic associations, we develop a style predictor using curated WikiArt corpus (WikiArt, 2025) of public-domain and permissively licensed images, organized into five broad, visually distinct style classes: *drawing*, *engraving*, *illustration*, *painting*, and *photography*. Although not exhaustive, these categories were selected because they capture distinct stylistic modes that emerged consistently in our initial qualitative analysis of historically themed generations, providing a structured foundation for identifying patterns of stylistic bias in generated imagery. The dataset consists of 1,280 training and 320 validation images per class (6,400 / 1,600 total). Since WikiArt does not distinguish between b&w and color photography, we apply a post-processing step using the colorfulness metric of Hasler & Suesstrunk (2003), re-labeling photographs with a score < 10 as *monochrome*.

We benchmark multiple encoders by training a linear probe on each frozen backbone, including CNNs (VGG-16, ResNet-50), vision transformers (Swin-Base, BEiT-Base, MAE ViT-Base, DINOv2 ViT-B/14), and a zero-shot CLIP baseline (ViT-B/32). DINOv2 (Oquab et al., 2023) achieves the strongest performance on the validation set (Acc. = 0.89, F1 = 0.87), outperforming all others, and is therefore adopted for all downstream analyses. Using this predictor, we assign a stylistic label to every *HistVis* generation, enabling systematic measurement of model-specific tendencies in historical representation. Detailed metrics and confusion matrices for all backbones are reported in Appendix D.

## 4.2 VISUAL STYLE DOMINANCE (VSD) SCORE

We define the *Visual Style Dominance* (VSD) score as the maximum proportion of images generated by model $m$ for period $t$ that are classified as a single style: $\text{VSD}(m, t) = \max_s P_m(s \mid t)$, where $P_m(s \mid t)$ is the proportion of images predicted as style $s$. Higher scores indicate strong convergence toward a dominant style; lower scores reflect greater stylistic diversity. To account for classification uncertainty, we estimate 95% confidence intervals (CI) via bootstrap resampling as detailed in Appendix F.

Table 1 shows VSD scores and dominant styles across models and time periods. In the 17th and 18th centuries, SDXL exhibits a strong preference for *engravings*, while SD3 and FLUX.1 favor *paintings*. In modern periods, FLUX.1 and SD3 converge on *photography*, whereas SDXL shifts toward *illustrations*. All three models display strong *monochrome* dominance in early 20th-century decades (1910s–1950s). Most VSD scores are statistically robust; however, for a few periods (e.g., FLUX.1 in the 1950s), overlapping CI preclude reliable identification of a single dominant style. While this may reflect classification uncertainty, it could also capture genuine cultural transitions, such as the shift from monochrome to color media during that period.

To assess mitigation potential, we applied prompt engineering, a common strategy for bias reduction in text and image generation (Clemmer et al., 2024; Dwivedi et al., 2023), and re-generated SDXL outputs using prompts that explicitly requested photorealism and discouraged monochrome aesthetics. These interventions occasionally reduced stylistic convergence (e.g., 17th–18th c.) or shifted the dominant style (e.g., from *monochrome* to *painting*), but often failed to override stylistic defaults, suggesting that such biases are persistent and not easily mitigated through prompting alone.

Table 1: VSD scores by model and historical period. Asterisks (*) denote cases where dominance was not statistically significant due to overlapping CI. The final columns indicate changes in SDXL after prompt engineering.

| Period | FLUX.1 | | SD3 | | SDXL | | SDXL → Mitigated | |
|--------|--------|--------------|-------|--------------|--------|--------------|-------|--------------|
| | Score | Visual Style | Score | Visual Style | Score | Visual Style | Score | Visual Style |
| 17th c. | 0.88 | painting | 0.86 | painting | 0.93 | engraving | 0.83 | engraving ↓↔ |
| 18th c. | 0.84 | painting | 0.79 | painting | 0.89 | engraving | 0.80 | engraving ↓↔ |
| 19th c. | 0.58 | painting | 0.64 | painting | 0.61 | engraving | 0.51 | engraving ↓↔ |
| 20th c. | 0.83 | photog. | 0.66 | photog. | 0.35* | illustr. | 0.82 | illustr. ↑↔ |
| 21st c. | 0.84 | photog. | 0.94 | photog. | 0.83 | illustr. | 0.90 | illustr. ↑↔ |
| 1910s | 0.75 | monochrome | 0.82 | monochrome | 0.73 | monochrome | 0.78 | painting ↑⇆ |
| 1930s | 0.77 | monochrome | 0.79 | monochrome | 0.73 | monochrome | 0.51 | painting ↓⇆ |
| 1950s | 0.50* | monochrome | 0.54 | monochrome | 0.66 | monochrome | 0.82 | illustr. ↑⇆ |
| 1970s | 0.91 | monochrome | 0.85 | photog. | 0.33* | photog. | 0.70 | illustr. ↑⇆ |
| 1990s | 0.95 | photog. | 0.96 | photog. | 0.50 | illustr. | 0.56 | illustr. ↑↔ |

**Legend:** ↑/↓: Score increase/decrease; ↔: Unchanged; ⇆: Style changed.

## 5 HISTORICAL CONSISTENCY

The second aspect of our benchmark evaluates historical consistency. This involves assessing whether TTI models introduce *anachronisms*, e.g. modern artifacts in the depiction of pre-modern contexts. Inspired by the open-set bias detection strategies proposed by D'Incà et al. (2024) and Chinchure et al. (2024), which identify biases without predefined categories and focus on cultural issues (e.g., gender stereotypes, brand associations), we apply a similar approach to target anachronisms and propose an automated detection methodology which we validate through human judgment.

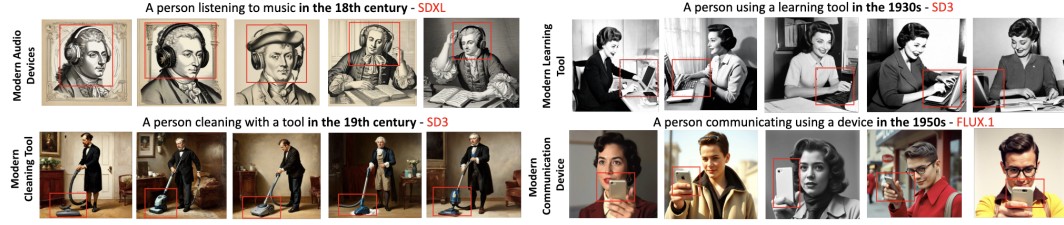

Figure 3: Examples of generated images with anachronisms identified by our two-stage method: headphones in the 18th, vacuum cleaner in the 19th-century, laptop in 1930s and smartphone in 1950s.

### 5.1 ANACHRONISM DETECTION

Our anachronism detection method is applied to the same structured prompts defined in *HistVis*, where each prompt is a combination of a human activity and a historical time period, denoted as $\langle a, t \rangle$. The detection pipeline consists of two stages: (i) LLM-guided Anachronism Proposal and (ii) VQA-based Anachronism Detection.

Given a prompt $\langle a, t \rangle$, an LLM is used to generate a set of potential anachronistic elements $\mathcal{Z}_{a,t} = \{z_1, \ldots, z_k\}$ in the historical context defined by $t$. For each proposed anachronism $z_i$, it also produces a corresponding binary yes/no question $q_i$ designed to identify whether the anachronism appears in the generated image. For example, given the prompt *A person listening to music in the 18th century,"*

the LLM might return the anachronism *"audio devices"* along with the identification question *"Is the person using audio devices, such as headphones or smartphones? Answer with 'yes' or 'no'.* For this stage, we evaluated multiple LLMs, including GPT-4o, LLaMA-3.2–11B, and Qwen2.5–7B. While GPT-4o generated the broadest and least redundant set of candidate anachronisms, LLaMA-3.2 achieved closely comparable coverage, underscoring that our framework is reproducible with open-source alternatives. Full comparisons across LLMs are provided in Appendix 5.

Given the generated image $I_{a,t}^m$ and the associated set of questions $\mathcal{Q}_{a,t} = \{(z_i, q_i)\}$, we query three VLMs, GPT-4o, LLaMA-3.2–11B, and Qwen2.5 VL–7B, in a VQA setup. Each model answers every question $q_i$ with a binary yes/no response, indicating whether the corresponding anachronistic element $z_i$ is visually present in the image. The final decision for each $q_i$ is obtained by majority vote across the three models, which reduces model-specific errors and improves agreement with human judgment compared to relying on a single VLM as detailed in Section 5.2.

**Anachronism Assessment Metrics** To systematically evaluate anachronisms in generated images, we first normalize lexical variations in the phrasing of proposed anachronisms introduced by the LLM. For instance, a model may refer to the same object as 'audio device' in one case and 'digital audio device' in another. To avoid double-counting these identical references, we apply *fuzzy matching* (SeatGeek, 2011) to treat them as the same entry. Importantly, this normalization only addresses surface-level naming inconsistencies and does not involve semantic grouping of distinct concepts.

For each time period $t \in \mathcal{T}$ and each TTI model $m \in \{\text{SDXL}, \text{SD3}, \text{FLUX.1}\}$, we compute two metrics for each anachronistic element $z_i$ proposed by the LLM. *Frequency* is defined as the proportion of generated images (for the given time period and model) in which $z_i$ is detected, and *Severity* as the consistency with which the model introduces $z_i$ once it is proposed by the LLM. Formally, we define:

$$\text{Frequency}(z_i, t, m) = \frac{n_{z_i}^{\text{detected}}}{N_t^m}, \quad \text{Severity}(z_i) = \frac{n_{z_i}^{\text{detected}}}{n_{z_i}^{\text{proposed}}}, \tag{1}$$

where $N_t^m$ is the total number of images generated by model $m$ for time period $t$, $n_{z_i}^{\text{detected}}$ is the number of images in which $z_i$ is detected, and $n_{z_i}^{\text{proposed}}$ is the number of prompts in which the LLM proposed $z_i$ as a plausible anachronism. For example, if "modern phone equipment" is detected in 10 out of 1,000 images for the 18th century, its frequency is 1%, and if it is proposed in exactly 10 instances and appears in all 10, then its severity is 1.0.

## 5.2 Anachronism Frequency & Severity Scores

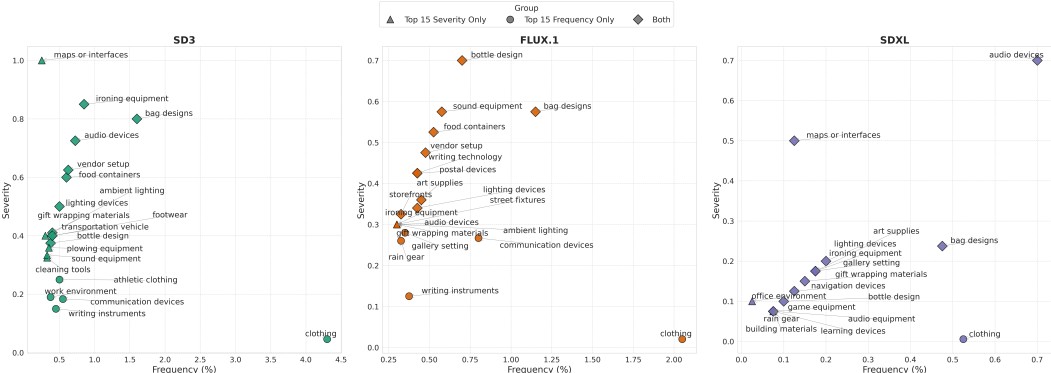

Figure 4: Top 15 anachronistic elements per model, ranked by frequency (x-axis) and severity (y-axis). Circles indicate elements in the top 15 by frequency, triangles by severity, and diamonds by both.

Figure 4 plots the top 15 anachronistic elements per model, ranked independently by *frequency* and *severity*, aggregated across all centuries. Each data point corresponds to an element proposed by the LLM and confirmed via VQA, with fuzzy matching applied to unify lexical variations. For example, *clothing* appears in about 2-5% of all anachronistic images, making it the most frequent item overall, yet it shows relatively low severity, indicating a scattered but recurring misrepresentation. By contrast, items like *audio devices* and *ironing equipment* occur less often but with higher severity: when these

elements are proposed by the LLM as plausible anachronisms for a given prompt, they are almost always identified in the corresponding generated images. This behavior suggests that models often rely more on the prompt's conceptual cues (e.g., "listening to music") than on the intended historical context, revealing a structural lapse in historical conditioning. Furthermore, S3 appears most prone to anachronisms, with 20% of its 19th-century and 25% of its 1930s images flagged, followed by FLUX.1 and SDXL, revealing varying model tendencies. Examples of high-severity anachronisms are shown in Figure 3, with detailed breakdowns in Appendix J.

**User Study.** To evaluate the alignment between our anachronism detection method and human judgment, we conducted an anonymous user study on a public crowdsourcing platform with no geographical restrictions. Since SD3 exhibited the highest anachronism rate, we sampled 200 images per decade and century from this model (1,800 in total). Each image was annotated by three participants, producing 5,400 responses, of which 2,040 high-quality annotations from 234 reliable participants were retained after filtering based on predefined control questions. Inter-annotator agreement, measured by Fleiss' $\kappa = 0.63$, indicated substantial agreement. Using the majority vote as the human baseline, our ensemble method reached 75% agreement with human judgments, substantially above chance, and outperformed individual VLMs (72% for GPT-4o, 68% for LLaMA-3.2, 63% for Qwen2.5). For context, prior work (D'Incà et al., 2024) reports 67% agreement on simpler contemporary VQA tasks (e.g., identifying the color of a car), suggesting that our approach performs competitively in a substantially richer historical and semantic setting. Appendix L provides additional details on the interface, instructions, and data collection.

## 6 DEMOGRAPHIC REPRESENTATION

The third dimension of our benchmark evaluates how TTI models portray gender and race in relation to LLM-predicted historically plausible demographics. The goal of this analysis is to reveal whether TTI models exhibit extreme patterns of over- and under-representation of certain demographic groups with respect to specific activities and time periods. Because our benchmark and dataset target broad period categories and general activities, a factual historical baseline grounded in real-world data across the full range of cultures, regions, periods and activities cannot be reliably established within the scope of this work. However, to still address historical demographic bias at least at a coarse level, we extract LLM-based estimates that synthesize contextual knowledge for each activity and period as starting points for analyzing over- and under-representation. These estimates are not substitutes for historical records or expert annotation, but serve as reference points for identifying major divergences, cases where model outputs reflect stereotypical assumptions about the prompt activity itself rather than the specified temporal condition. While this approach provides a practical and scalable baseline, it remains a coarse approximation of historical accuracy and inherits the limitations and biases of the underlying language model.

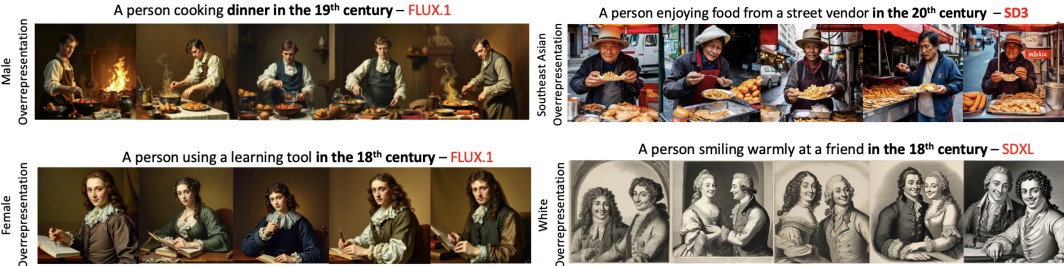

Figure 5: Examples of generated images illustrating demographic overrepresentation across models, time periods, and activities.

### 6.1 METHODOLOGY

To examine demographic representation in historically conditioned TTI images, we implement a three-step pipeline that compares generated demographics to LLM-predicted historically plausible expectations: (1) extracting attributes from generated images with an automated face classifier, (2)

estimating expected distributions with an LLM that synthesizes contextual knowledge for each activity–time pair, and (3) quantifying under- and overrepresentation through metric-based comparison.

**Demographic Extraction from Generated Images.** We use the FairFace classifier (Karkkainen & Joo, 2021), a ResNet34-based model trained on 108,501 facial images with balanced demographic attributes, to detect gender (male, female) and race (White, Black, Indian, East Asian, Southeast Asian, Middle Eastern, Latino) in generated images, including those with multiple faces. For each image $I_{(a,t)}$ corresponding to an activity–period pair $\langle a, t \rangle$, the classifier outputs $\{(g_j, r_j, c_j^{\text{gender}}, c_j^{\text{race}})\}_{j=1}^n$, where $g_j$ and $r_j$ denote the gender and race of the $j$th face and $c_j^{\text{gender}}, c_j^{\text{race}} \in [0, 1]$ are confidence scores. Predictions below 0.7 in either attribute are discarded to reduce misclassification. Demographic proportions are computed by normalizing valid predictions across all detected faces in each image and then averaging over the $N$ images for each pair $\langle a, t \rangle$, yielding the distributions $P_{\text{gender}}^m(a, t)$ and $P_{\text{race}}^m(a, t)$.

To assess reliability, we cross-validate predictions using the DeepFace classifier (Serengil & Ozpinar, 2020), an ensemble of VGG-Face (Parkhi et al., 2015), FaceNet (Schroff et al., 2015), DeepFace (Taigman et al., 2014), and ArcFace (Deng et al., 2019). On 5,000 randomly sampled images (500 per period–model pair), we observe strong inter-model agreement: 95.9% for gender ($\kappa = 0.93$) and 90.8% for race ($\kappa = 0.83$), where $\kappa$ denotes Cohen's kappa coefficient for inter-rater reliability.

**Estimating Historical Demographic Baselines.** To generate estimates of historically plausible expectations, we use LLM-based predictions conditioned on each activity–period pair $\langle a, t \rangle$. For each case, the model is prompted with: *"Given the [activity] and the [historical period], estimate the plausible demographic breakdown (in percentages) for gender and race."* This yields baseline distributions $\hat{P}_{\text{gender}}^{\text{llm}}(a, t)$ and $\hat{P}_{\text{race}}^{\text{llm}}(a, t)$. Although real-world data is not available for the full scope of our categories and periods, we partially assess the LLM-based estimates by comparing them with the closest available data for three categories. We validated four candidate LLMs, GPT-4o, Claude 3.5 Sonnet, LLaMA-3.2–11B, Qwen2.5 VL–7B, against Our World in Data (OWID) statistics on gender and continent-level trends, focusing on categories with coverage since the 19th century: *Education* (Ritchie et al., 2023), *Agriculture* (Roser, 2023), and *Work & Collaboration* (Ortiz-Ospina & Roser, 2023). Because OWID does not provide race-based statistics and to avoid oversimplified region–race mappings, this evaluation of LLMs replaces race estimation with continent-level estimation. Our main analyses compare LLM race estimates with predictions derived from the FairFace classifier.

GPT-4o achieved the lowest MAE (4.64), narrowly ahead of LLaMA-3.2 (4.83), while Claude 3.5 Sonnet (6.93) and Qwen2.5 VL–7B (7.66) performed less consistently. We therefore adopt GPT-4o as our primary baseline, while recognizing LLaMA-3.2 as a strong open-source alternative. Appendix O further reports full category-level deviations with both backbones (Tables 16, 17), confirming that results are robust across estimators.

**Under & Overrepresentation Metric.** We propose two metrics, *underrepresentation* and *overrepresentation*, to quantify how closely TTI-generated images align with the estimated historically plausible demographic expectations. Inspired by fairness metrics in ML that quantify deviations from reference distribution (Wang et al., 2022; Hall et al., 2023b; Bianchi et al., 2023), our approach adapts these to the historical domain using LLM-derived estimates rather than population statistics. For each activity–period pair $\langle a, t \rangle$ and demographic category $d \in \{\text{male}, \text{female}, \text{White}, \text{Black}, \text{Middle Eastern}, \text{Indian}, \text{Southeast Asian}, \text{East Asian}\}$, we compute both metrics. Formally, let $P_d^{\text{model}}$ denote the proportion of demographic group $d$ observed in the generated images, and $\hat{P}_d^{\text{llm}}$ the corresponding estimate provided by the LLM. We define:

$$\text{Under}_d = \mathbb{I}\{P_d^{\text{model}} < \hat{P}_d^{\text{llm}}\} \cdot \left[\hat{P}_d^{\text{llm}} - P_d^{\text{model}}\right], \tag{2}$$

$$\text{Over}_d = \mathbb{I}\{P_d^{\text{model}} > \hat{P}_d^{\text{llm}}\} \cdot \left[P_d^{\text{model}} - \hat{P}_d^{\text{llm}}\right] \tag{3}$$

These two metrics measure the absolute deviation in cases where the TTI model underrepresents or overrepresents a demographic group relative to the LLM-derived historically plausible baseline. We compute these values for each prompt and then aggregate them at the domain category level (e.g., education, arts) to identify systematic demographic disparities across broader cultural contexts.

## 6.2 DEMOGRAPHIC ALIGNMENT EVALUATION

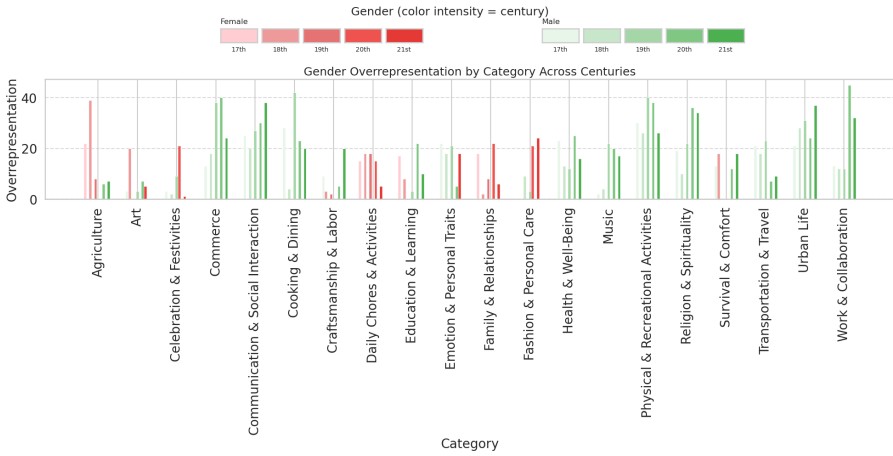

(a) Gender overrepresentation across the 17th–21st centuries.

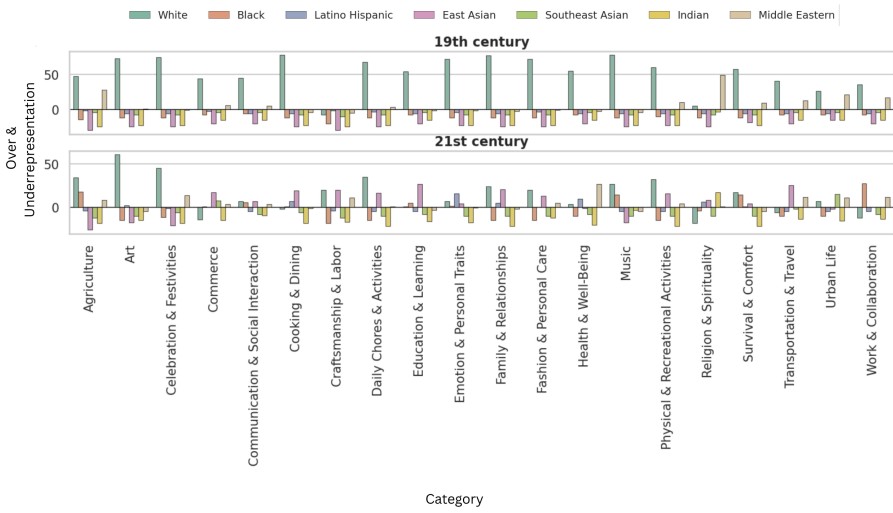

(b) Racial under- and overrepresentation in the 19th and 21st century.

Figure 6: Gender and racial over-/underrepresentation in FLUX.1 outputs across different centuries and activities, measured as absolute deviations from GPT-4o demographic baselines.

When comparing GPT-4o-derived demographic baselines with face classifier predictions across TTI models, we find frequent divergences from the estimated historically plausible patterns, with variation depending on the model. As shown in Figure 6a, FLUX.1 tends to overrepresent male figures, even in categories like "cooking & dining," where GPT-4o estimates a predominantly female presence until the 20th century. This male bias persists across centuries, including in "work & collaboration," which GPT-4o anticipates becoming gender-balanced by the 21st century. Conversely, "education & learning" skews female in the 17th–18th centuries, although GPT-4o indicates male majorities for that period. White individuals are generally overrepresented across models, though this trend diminishes by the 21st century, aligning closer to LLM-estimated historical expectations. However, specific categories (e.g., "religion & spirituality," "agriculture") show pronounced spikes in non-White representations, as illustrated in Figure 6b for FLUX.1. Similar trends are visible across other models and time periods, suggesting that these distributions may reflect correlations in the training data rather than historically grounded depictions. Additional figures with representation scores across all models, time periods, and activity categories are provided in Section Q.

## 7 DISCUSSION

### 7.1 LIMITATIONS

While our study offers the first systematic evaluation of historical representation in TTI models, it is important to emphasize several key limitations. First, our analysis of stylistic defaults is based on broad categories and therefore does not capture finer stylistic distinctions, which remains a potential direction for future work . Second, although we selected a range of different centuries, as well as decades, to capture broad trends and subtler shifts, these choices were somewhat arbitrary; alternative periods could have been chosen and might have yielded different insights. Third, in our benchmark we decided to focus on a broad spectrum of universal activities for this initial investigation, and we are aware that this may omit details relevant to specific historical contexts. Fourth, while our use of LLMs for demographic estimation and anachronism detection demonstrates promising initial results, we remain mindful of the challenges inherent in historical analysis, the potential biases highlighted in prior research (Gallegos et al., 2024; Navigli et al., 2023), and the fact that LLMs cannot replace the depth or reliability of expert historical knowledge. We are also aware that our baselines are significantly simplified, as they are obtained by averaging demographic expectations across period and activity and therefore omit other potentially relevant factors. Similarly, the use of face classifiers further oversimplifies social complexities by reducing gender and race to discrete categories. Finally, while our anachronism pipeline was validated with a user study, judgments may vary due to differences in historical expertise and subjective interpretation. Moreover, the LLM-generated anachronism list is not exhaustive; it targets prominent and recurring inconsistencies rather than every possible historical deviation. Despite these limitations, our approach establishes a strong inital basis for systematically evaluating historical aspects of TTI models, and its flexibility allows application to future models for continuous improvement.

### 7.2 HISTORICAL PLAUSIBILITY AND DIVERSITY

Evaluating demographic patterns in historically-situated synthetic imagery raises an inherent ethical dilemma: how to generate content that accurately depicts the past while avoiding that such content further propagates biases and reinforces exclusionary narratives. Recent work on sociopolitical aspects of generated images (Shaw et al., 2025) suggests that users are more receptive to diversity when it is transparently and authentically contextualized, indicating that neither strict historical fidelity nor unrestricted diversification is sufficient on its own. Striking a balance between these competing aims remains an open challenge. Within this context, we find it necessary to emphasize that we position our LLM-derived historical plausibility as one possible evaluation axis, complementary to others such as cultural sensitivity, aesthetic quality, and faithfulness to the prompt, and not as an expected standard. Our intention is not to prescribe a "correct" way of depicting the past, but to provide a starting point for addressing this challenge at a broader, systematic scale.

## 8 CONCLUSION

We presented a benchmark for assessing how TTI diffusion models represent historical contexts. Our analysis is focused on three aspects: stylistic bias, chronological inconsistencies and demographic representations. Our results suggest that current models rely on narrow and stereotyped stylistic cues rather than nuanced understandings of historical periods, often defaulting to era-specific stylistic stereotypes and producing photorealistic depictions only from the 20th century onward. In addition, frequent anachronisms suggest that historical periods are not cleanly separated in the latent spaces of these models, since modern artifacts often emerge in pre-modern settings, undermining the reliability of TTI systems in education and cultural heritage contexts. Finally, our demographic analysis indicates that TTI outputs often diverge from LLM-estimated historically plausible expectations, sometimes aligning with contemporary expectations and sometimes reflecting activity-related biases rather than the specified historical condition. Ultimately, balancing historical fidelity with broader representational goals remains an open challenge, and future work should integrate richer historical knowledge and interdisciplinary expertise to develop models that depict the past with greater accuracy and sensitivity.

## 9 ACKNOWLEDGMENTS

This research was supported by the Swiss National Science Foundation (SNSF) under the Ambizione Grant Scheme, Grant No. 216104.

## 10 ETHICS STATEMENT

**Human Subjects**  Our study includes human subjects for the purpose of evaluating anachronism detection. All annotations were carried out by adult participants recruited under fair working conditions and compensated at or above the local minimum wage. The task was limited to visual evaluation of synthetic images, and did not involve collection of personally identifiable information (PII) or sensitive data. Participants provided informed consent and were free to withdraw at any time.

**Data Sources**  The image data used in training and evaluation are sourced from WikiArt, which contains both public domain artworks and copyright-protected artworks that it hosts under the U.S. "fair use" principle. In line with legal and ethical standards, we filtered the dataset to include only artworks that are explicitly in the public domain or released under a license permitting research use. No copyrighted or private data were used without permission.

**Demographic Classification**  For demographic analysis, we employed a face classifier (FairFace) that treats race and gender as categorical attributes. We recognize that such classifications simplify complex, socially constructed identities and may reproduce existing biases. We therefore report demographic results in aggregate, and interpret them cautiously, emphasizing limitations and potential sources of misclassification. Our intention is not to essentialize or reify these categories, but to critically assess how generative models encode and reproduce demographic representation.

## 11 REPRODUCIBILITY STATEMENT

Both the dataset[1], including all associated metadata, and our reproducible evaluation scripts[2] are publicly available under cc by-nc 4.0 license.

In Appendix B, we provide the full prompt design used in the HistVis dataset, with all prompts organized by category. This structure is intended to support future applications and benchmarking across different TTI models.

Image generation was performed using an NVIDIA A100 GPU (40GB), with 24 CPU cores, 64GB RAM, and Python 3.12.1.

Some of the models used for evaluation (e.g., GPT-4o) are proprietary and accessed via API, so we cannot guarantee that future queries will yield identical outputs. To support reproducibility, we also run our analyses using open-source alternatives and report the strongest substitutes in Appendix H (LLM Backbones for Anachronism Proposal), Appendix O (Validation of Demographic Estimation with Historical Data), and Appendix P (Comparative Results for Demographic Deviations with an Open-Source Alternative).

We provide all human annotation instructions as screenshots of the task interface, exactly as seen by participants, in Appendix K.

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

# A    LARGE LANGUAGE MODEL USAGE

We used a large language model (LLM) as a writing assistant for polishing grammar, improving clarity, and rephrasing some of text included in our paper.

# B    HISTVIS PROMPT DESIGN

The HistVis dataset uses *universal activity prompts* (e.g., "a person working in the 18th century") rather than historically specific events. This choice was guided by the following considerations:

- **Minimizing prompt bias**
  Prompts are deliberately simple and neutral to avoid encoding historical facts, stylistic cues, or demographic priors. This allows us to observe how models default to particular representations.
  *Example:* The prompt "a person cooking in the 18th century" leads some models to generate male-presenting figures, revealing latent gender bias not specified in the input.

- **Cross-temporal comparability**
  All 100 activity prompts are reused across 10 historical periods, enabling consistent comparisons. Activities are conceptually universal (e.g., working, celebrating, communicating), even if their historical realizations differ. This isolates the effect of temporal conditioning from prompt content.

**Prompt construction.**    All prompts follow a predefined template (*"a person [activity] in the [historical period]"*), and the set of time periods was specified by the authors in advance. LLMs were used during the early ideation stage to surface candidate activities, which was then manually curated by the authors.

**Limitations.**    While our prompts support consistency and neutrality, they do not capture historically grounded facts that could reveal additional gaps in models' historical knowledge. Future work could extend this approach with historically specific events to assess how models respond to more context-rich and accurate historical inputs.

**HistVis Prompts.**    Below are all the prompts used in the HistVis dataset, organized by category.

---

**Music**

1. A person listening to music
2. A person performing music
3. A person playing an instrument
4. A person listening to a live performance
5. A person singing

---

**Art**

6. A person creating art
7. A person using colors to create something
8. A person admiring an artwork
9. A person sharing their artwork
10. A person sketching something

---

### Communication and Social Interaction

11. A person communicating using a device
12. A person negotiating with someone
13. A person sending a message
14. A person listening to someone speak
15. A person spreading a rumor

### Family and Relationships

16. A person spending time with their family
17. A person laughing with a friend
18. A person holding hands with a loved one
19. A person embracing their child
20. A person welcoming a loved one home

### Transportation and Travel

21. A person traveling from one place to another
22. A person waiting for a ride
23. A person carrying belongings during a journey
24. A person navigating using a device
25. A person preparing for a journey

### Urban Life

26. A person carrying shopping bags
27. A person walking through a market
28. A person navigating through a street
29. A person waiting at a public gathering spot
30. A person enjoying food from a street vendor

### Physical and Recreational Activities

31. A person dancing with someone
32. A person running
33. A person playing a game with someone
34. A person discovering a new toy
35. A person exercising

### Cooking and Dining

36. A person sharing a meal with another
37. A person cooking dinner
38. A person setting a table
39. A person drinking from a bottle
40. A person holding a bowl of food

### Daily Chores and Activities

41. A person cleaning with a tool
42. A person washing laundry
43. A person organizing items in a bag
44. A person feeding an animal
45. A person ironing clothes

### Craftsmanship and Labor

46. A person observing a building under construction
47. A person repairing an object
48. A person making clothes
49. A person building a shelter
50. A person painting a surface

### Religion and Spirituality

51. A person praying
52. A person celebrating a religious event
53. A person reading a sacred text
54. A person lighting a candle in a place of worship
55. A person meditating quietly

### Celebration and Festivities

56. A person unwrapping a gift
57. A person dancing during a festival
58. A person toasting with a glass at a celebration
59. A person cheering at a celebration
60. A person decorating a space for a celebration

### Survival and Comfort

61. A person using a tool to illuminate a dark space
62. A person lighting a fire
63. A person gathering water from a natural source
64. A person shielding themselves from rain
65. A person shielding their eyes from the sun

### Health and Well-being

66. A person taking medicine
67. A person examining a patient
68. A person assisting someone who is injured
69. A person treating a wound with care
70. A person applying a medical treatment

### Education and Learning

71. A person studying
72. A person teaching music to another
73. A person guiding someone in understanding
74. A person using a learning tool
75. A person reading and taking notes

### Agriculture

76. A person planting seeds in the ground
77. A person tending a vegetable garden
78. A person caring for livestock
79. A person watering crops
80. A person plowing a field

### Fashion and Personal Care

81. A person putting on clothing
82. A person showing off their style
83. A person accessorizing their outfit
84. A person shaving their face
85. A person combing their hair

---

**Emotion and Personal Traits**

    86. A person showing bravery

    87. A person expressing joy after achieving a goal

    88. A person smiling warmly at a friend

    89. A person expressing gratitude to another

    90. A person standing up for a cause

---

**Work and Collaboration**

    91. A person working

    92. A person organizing their work equipment

    93. A person reviewing a group's progress on a work-related task

    94. A person assisting a coworker with a challenge

    95. A person preparing for a day of work

---

**Commerce**

    96. A person selling items at a market stall

    97. A person bargaining with a vendor

    98. A person wrapping goods for a customer

    99. A person opening a shop

    100. A person closing a register after a sale

## C  WHY STYLISTIC BIAS CAN BE A PROBLEM

While stylistic bias does not introduce factual errors, it raises critical issues about cultural stereotypes and visual assumptions embedded in generative models. Prior work (Senthilkumar et al., 2024) shows that models often default to cartoonish depictions for non-Western artifacts, lacking cultural nuance. We observed a similar pattern: when prompted for "photorealistic" images from earlier centuries, models often reverted to engraving-like or monochrome styles. Even when directly instructed, models frequently failed to override these defaults, indicating that historical periods are often visually "hard-coded" based on past visual documentation conventions. This undermines the flexibility of style as a controllable attribute and can reinforce learned stereotypes about what the past is "supposed" to look like.

## D  STYLE CLASSIFIER: TRAINING & FULL REPORTS

**Training Configuration**  We benchmarked seven encoders by training a linear classifier on frozen features extracted from a curated WikiArt-derived dataset comprising 6,400 training and 1,600 validation images across five stylistic categories. All images were resized to $224 \times 224$ pixels and normalized to the $[0, 1]$ range by scaling pixel values by $1/255$. To enhance generalization, we applied standard data augmentation, including random rotations (up to $20°$), translations (10%), shear and zoom transformations (10%), and horizontal flipping. Training was performed using a linear probe setup (frozen encoder with a trainable linear head), optimized with cross-entropy loss and the Adam optimizer (learning rate $1 \times 10^{-3}$), using a batch size of 32 for up to 50 epochs. All models were trained on a single NVIDIA A100 GPU.

In parallel, we evaluated CLIP ViT-B/32 in a zero-shot setting. Images were classified based on cosine similarity to handcrafted textual prompts representing each style category (e.g., "an image of a painting"). We experimented with several prompt structures and report results for the best-performing variant. Despite its broad representational capacity, CLIP underperforms on fine-grained

style distinctions (Macro F1 = 0.66), particularly for less semantically grounded classes such as *drawing* and *illustration*. Based on its superior overall and per-class performance (Macro F1 = 0.88), we selected DINOv2 ViT-B/14 as the default encoder for all downstream analyses.

**Classification Report** Table 2 presents the classification performance of each encoder on the WikiArt-derived validation set. We report overall accuracy, macro-averaged F1 score, and per-class F1 scores for the six style categories. Among the supervised models, DINOv2 ViT-B/14 achieves the highest macro F1 score (0.876) and ties with Swin-B for best overall accuracy (0.896), demonstrating strong and balanced performance across all classes. In contrast, the zero-shot CLIP baseline shows competitive results for *painting* and *photography* but struggles with less semantically grounded styles such as *drawing* and *illustration*, leading to a lower macro F1 score of 0.658. These results validate our choice of DINOv2 as the default encoder for downstream stylistic analyses.

Table 2: Classification performance of different encoders on the WikiArt-derived dataset. Metrics are reported as F1-scores per class, with overall accuracy and macro average F1.

| Model | Accuracy | Macro F1 | Drawing | Engraving | Illustration | Painting | Photography |
|---|---|---|---|---|---|---|---|
| CLIP ViT-B/32 | 0.734 | 0.658 | 0.488 | 0.539 | 0.436 | 0.905 | 0.925 |
| ResNet-50 | 0.879 | 0.852 | 0.783 | 0.788 | 0.826 | 0.965 | 0.900 |
| VGG-16-BN | 0.858 | 0.827 | 0.746 | 0.743 | 0.804 | 0.951 | 0.892 |
| Swin-B | 0.896 | 0.868 | 0.825 | 0.797 | 0.828 | 0.974 | 0.916 |
| BEiT-Base | 0.888 | 0.857 | 0.813 | 0.788 | 0.796 | 0.976 | 0.912 |
| MAE ViT-Base | 0.873 | 0.848 | 0.780 | 0.788 | 0.826 | 0.951 | 0.894 |
| DINOv2 ViT-B/14 | 0.896 | **0.876** | 0.802 | 0.850 | 0.836 | 0.957 | 0.935 |

# E PREDICTED VISUAL STYLES BEFORE AND AFTER MITIGATION ATTEMPT

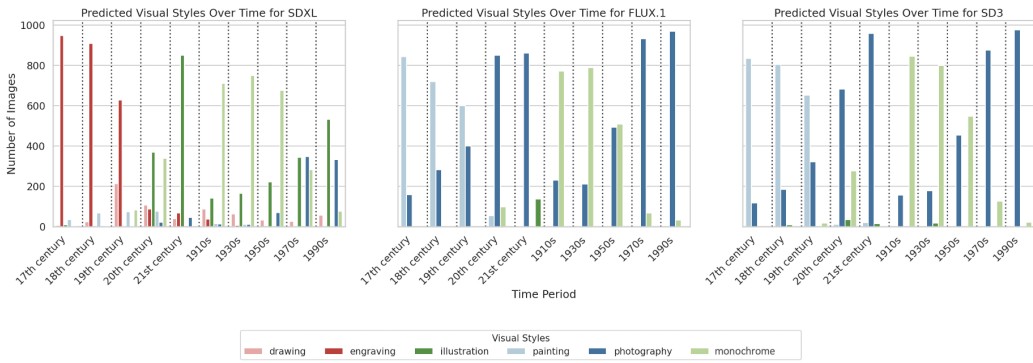

Figure 7: Predicted visual styles of generated images across different historical periods for each TTI model using DINOv2. Each historical period is evaluated using 1,000 generated images per model.

Figure 7 presents the predicted stylistic properties of generated images across time periods $\mathcal{T}$ for each TTI model using DINOv2. Even though no explicit stylistic instructions were provided in the prompts, models exhibit consistent yet model-specific stylistic tendencies. For instance, SDXL predominantly generates engraving-like images for the 17th and 18th centuries (e.g., 947 of 1,000 images for the 17th century, 909 for the 18th), while SD3 and FLUX.1 favor painting-like renderings for these same periods. In the 20th and 21st centuries, SD3 and FLUX.1 predominantly generate photography-like images, whereas SDXL exhibits greater stylistic diversity, generating illustrations, drawings, engravings and some photography. More specifically, in the 1910s, 1930s, and 1950s, SD3 and FLUX.1 mostly generate b&w photographs, reinforcing the historical association of these decades with monochrome photography. In contrast, SDXL produces a wider variety of styles, including illustrations and engravings, even when generating images for the same time periods. These results indicate that TTI models encode implicit associations between historical periods and specific stylistic renderings, which emerge even when no style is specified in the prompt, raising important

questions about how generative models learn and internalize visual conventions from their training data.

**Visual Bias Mitigation**   Prompt engineering has been explored as a bias mitigation strategy in both text and image generation, with studies demonstrating its effectiveness in reducing demographic biases in LLMs and guiding TTI models toward more balanced outputs Clemmer et al. (2024); Dwivedi et al. (2023). Taking that into consideration, we examined whether the implicit stylistic tendencies of SDXL, which exhibited the most diverse associations across historical periods (e.g., engravings for the 17th and 18th centuries, drawings for the 19th century, and illustrations for the 21st century), could be adjusted through targeted prompt engineering. Specifically, we replaced the standard neutral prompt (e.g., "A person [activity] in the [time period]") with a more explicit instruction ("A photorealistic image of [activity] in the [historical period]"). Additionally, we introduced a negative prompt ("black and white image") to discourage monochrome outputs by applying negative weights, a common practice in diffusion-based TTI models to steer the generation process away from unwanted elements Armandpour et al. (2023). The effectiveness of these modifications was then evaluated using DINOv2 to predict the visual styles of the newly generated images.

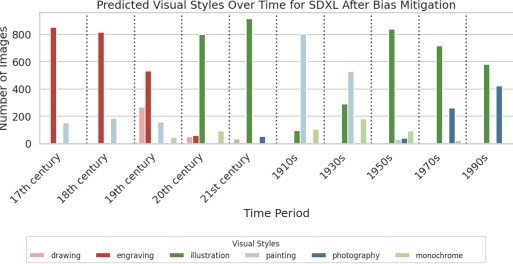

Figure 8: Predicted visual styles of SDXL-generated images across different historical periods after applying bias mitigation via prompt engineering. Each historical period is evaluated using 1,000 generated images.

Figure 8 shows the visual predictions of SDXL images after applying bias mitigation through targeted prompt engineering. Even when explicitly prompting for "photorealistic images," the distribution of stylistic outputs at the century level remains largely unchanged. SDXL continues to predominantly generate engraving-like images in the 17th, 18th and 19th century, suggesting that its deep-rooted historical stylistic associations are not easily overridden by prompt modifications alone. At the decade level, however, some shifts are observed. Notably, the proportion of monochrome images decreases following bias mitigation. Yet, rather than transitioning toward color photography as one might expect, SDXL instead increases the generation of paintings and illustrations. These results indicate that while prompt-level interventions can partially influence stylistic biases at a more granular level, such as reducing the proportion of monochrome outputs, they remain insufficient to fully counteract the model's learned stylistic tendencies. Figures 9 and 10 illustrate how certain visual styles either persist or shift following the application of the bias mitigation approach.

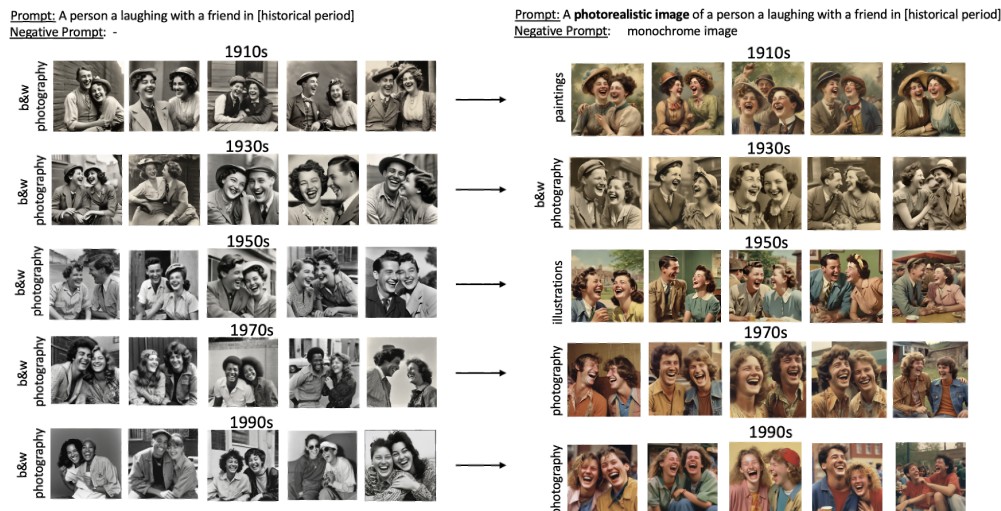

Figure 9: Predicted visual styles of generated images derived from the prompt "A person laughing with a friend in the [historical period]" (left) and "A photorealistic image of a person laughing with a friend in the [historical period]" (right) with "monochrome picture" used as a negative prompt.

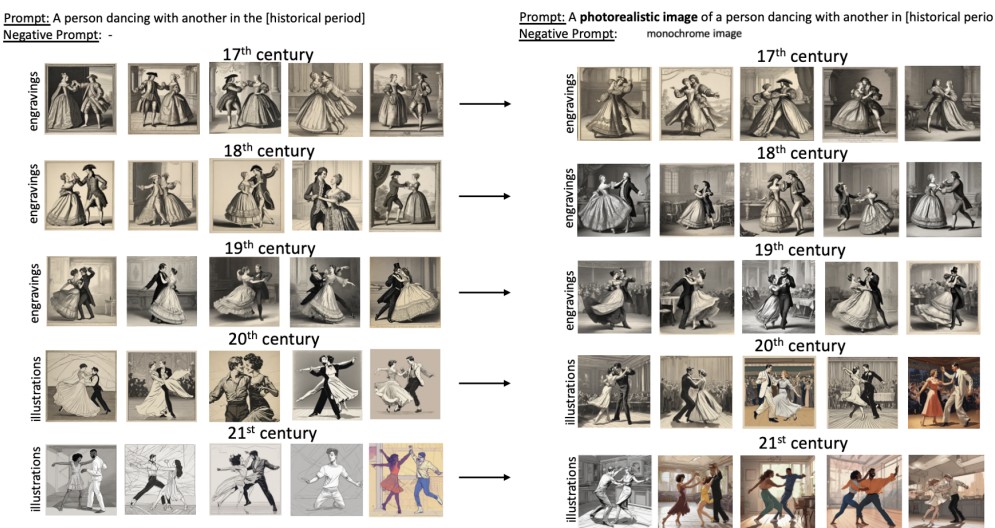

Figure 10: Predicted visual styles of generated images derived from the prompt "A person dancing with another in the [historical period]" (left) and "A photorealistic image of a person dancing with another in the [historical period]" (right) with "monochrome picture" used as a negative prompt.

## F  VSD: CONFIDENCE INTERVALS & ROBUSTNESS

To assess the robustness of our Visual Style Dominance (VSD) scores, we implemented a bootstrapping procedure that quantifies uncertainty arising from classifier error. For each model–period pair, we generated 5,000 bootstrap replicates by resampling with replacement from the corresponding image set. To account for misclassification noise, each predicted label was retained or reassigned according to the class-specific precision of DINOv2 (e.g., a "drawing" prediction with 0.80 precision was retained 80% of the time). For each bootstrap replicate, we recomputed the style proportions and corresponding VSD score.

This procedure yields empirical 95% confidence intervals (CIs) for the dominant style's share as well as for the second-most frequent style. The CIs therefore indicate how stable style dominance is under

Table 3: VSD scores with 95% confidence intervals and dominant/second styles for FLUX.1, SD3, and SDXL. Asterisk (*) indicates dominance is not statistically significant (overlapping CIs).

| Model | Period | VSD (95% CI) | Dominant Style | Second Style (Prop, 95% CI) |
|---|---|---|---|---|
| FLUX.1 | 1950s | 0.47 (0.46, 0.51)* | monochrome | photography   0.46 (0.42, 0.49) |
| FLUX.1 | 19th c. | 0.57 (0.54, 0.60) | painting | photography   0.38 (0.35, 0.41) |
| SD3 | 1950s | 0.51 (0.47, 0.54) | monochrome | photography   0.42 (0.39, 0.45) |
| SD3 | 19th c. | 0.63 (0.60, 0.66) | painting | color_photography   0.31 (0.28, 0.34) |
| SD3 | 20th c. | 0.63 (0.60, 0.66) | photography | monochrome   0.24 (0.22, 0.27) |
| SDXL | 1910s | 0.69 (0.66, 0.72) | monochrome | illustrations   0.14 (0.12, 0.16) |
| SDXL | 1930s | 0.70 (0.67, 0.73) | monochrome | illustrations   0.16 (0.14, 0.18) |
| SDXL | 1950s | 0.63 (0.60, 0.66) | monochrome | illustrations   0.19 (0.17, 0.21) |
| SDXL | 1970s | 0.33 (0.31, 0.36)* | photography | illustrations   0.30 (0.27, 0.33) |
| SDXL | 1990s | 0.44 (0.41, 0.47) | illustrations | photography   0.37 (0.34, 0.40) |
| SDXL | 19th c. | 0.54 (0.51, 0.57) | engraving | drawings   0.19 (0.17, 0.22) |
| SDXL | 20th c. | 0.33 (0.32, 0.36)* | monochrome | illustrations   0.32 (0.29, 0.35) |

Table 4: VSD scores with 95% confidence intervals and dominant/second styles for SDXL (Mitigated).

| Period | VSD (95% CI) | Dominant Style | Second Style (Prop, 95% CI) |
|---|---|---|---|
| 1930s | 0.51 (0.48, 0.54) | painting | illustrations   0.26 (0.23, 0.29) |
| 1950s | 0.70 (0.67, 0.72) | illustrations | monochrome   0.09 (0.07, 0.10) |
| 1970s | 0.61 (0.57, 0.64) | illustrations | photography   0.23 (0.21, 0.26) |
| 1990s | 0.49 (0.46, 0.52) | illustrations | photography   0.42 (0.39, 0.45) |
| 19th c. | 0.46 (0.43, 0.49) | engraving | drawings   0.21 (0.18, 0.23) |

classifier uncertainty. In most cases, the intervals are well separated, supporting that the reported dominant style is statistically robust. In a few cases, however, the intervals overlap, which implies that multiple styles are plausible candidates for dominance.

For example, in FLUX.1 (1950s) the intervals for monochrome (0.46–0.51) and color photography (0.42–0.49) substantially overlap, reflecting the historical transition from monochrome to color photography in this period. Similarly, SDXL in the 1970s (color photography 0.31–0.36 vs. illustrations 0.27–0.33) and SDXL in the 20th century (monochrome 0.32–0.36 vs. illustrations 0.29–0.35) exhibit overlapping intervals. We mark such cases with an asterisk in the summary tables.

## G   ADDITIONAL QUALITATIVE EXAMPLES OF STYLISTIC BIAS

The examples illustratd in Figure 11 illustrate how TTI models implicitly associate specific stylistic patterns with historical periods. Rather than generating images solely based on the historical context in the prompt, they impose learned stylistic conventions, even when no explicit artistic style is specified.

- **A person showing off their style in the**:
    - **17th Century** - Paintings (SD3)
    - **18th Century** – Engravings (SDXL)
    - **1990s** – Photography (FLUX.1)
- **A person teaching music to another in the**:
    - **19th Century** – Paintings (SD3)
    - **21st Century** – Illustrations (SDXL)
    - **1910s** - Monochrome Photography (FLUX.1)
- **A person embracing their child in the**:
    - **19th Century** – Drawings (SDXL)
    - **19th Century** - Paintings (SD3)

– **20th Century** - Monochrome Photography (FLUX.1)

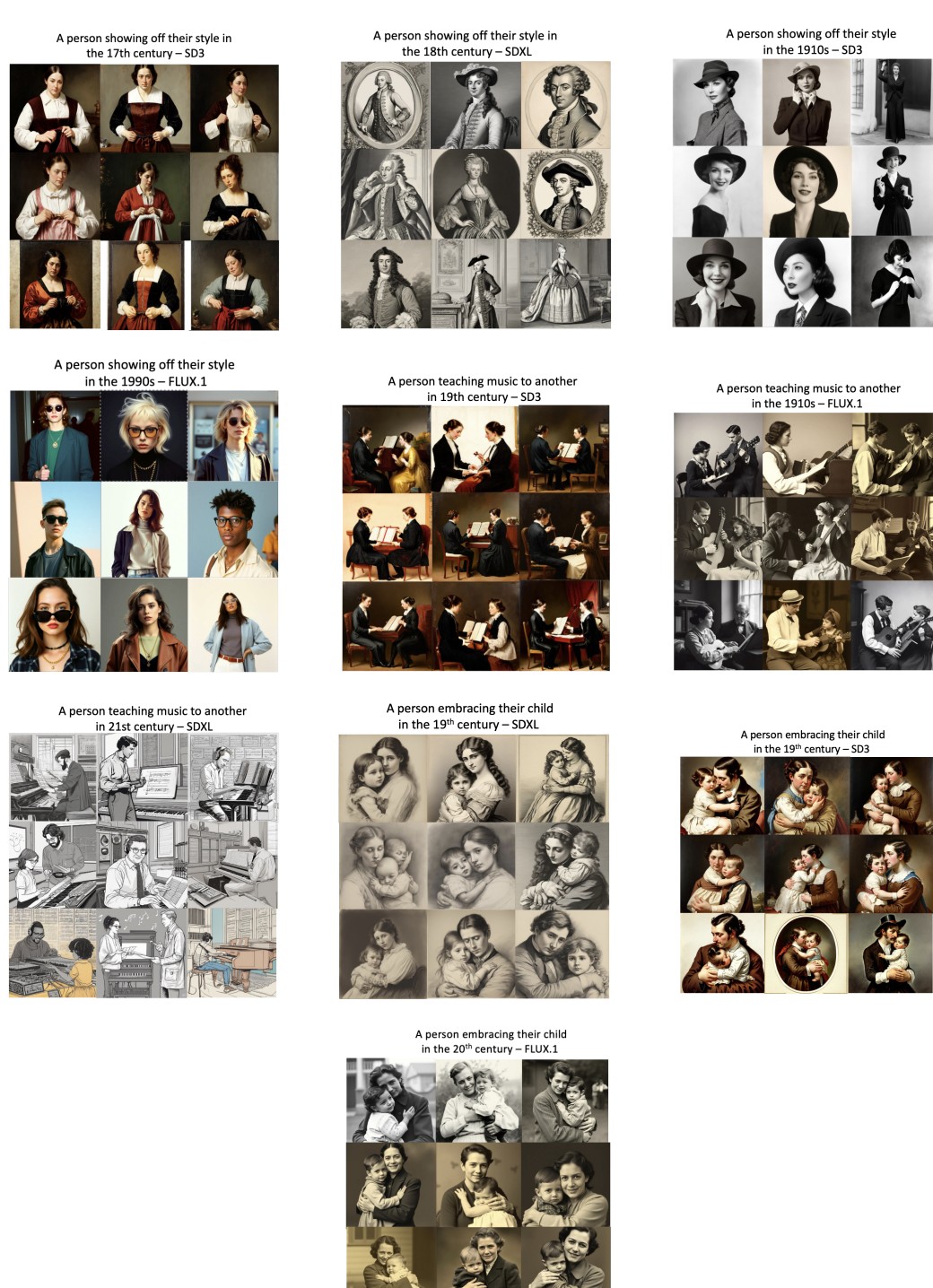

Figure 11: Examples of visual outputs across styles, activities, and time periods.

## H  ANACHRONISM PROPOSAL

A key step in our evaluation pipeline is to generate a structured set of potential anachronisms for each activity–period prompt. Rather than relying on predefined object lists, we frame this as an open-ended proposal task: an LLM is asked to anticipate which historically implausible artifacts might plausibly appear in a generated image. Each proposed artifact is then converted into an identification question that can be applied to the images produced by TTI models.

**Anachronism Identification:** You will be provided with a list of prompts describing people engaged in specific activities during various historical time periods. These prompts will serve as input for a Text-to-Image Generative Model like Stable Diffusion. Based on each prompt, perform the following tasks:

- Identify potential anachronisms that might appear in the generated image. Ensure that the list is relevant to the activity, time period, and setting described in the prompt.

- For each identified anachronism, generate a question to determine whether the anachronism appears in the generated image. Each question should end with: *'Answer with 'yes' (if the anachronism is present) or 'no' (if it is absent)."*

All answers must be in json format.

**Example JSON Representation:**

Listing 1: Example JSON structure for anachronism detection

```
1  {
2    "index": 1,
3    "prompt": "A person listening to music in the 19th century",
4    "possible_anachronisms": [
5      "modern music devices",
6      "modern clothing",
7      "modern surroundings"
8    ],
9    "questions_to_identify_anachronisms": {
10     "modern music devices": "Is the person using modern music devices, such as headphones or
          smartphones that didn't exist in the 19th century? Answer with 'yes' or 'no'.",
11     "modern clothing": "Is the person wearing modern clothing styles not typical for the 19th
          century? Answer with 'yes' or 'no'.",
12     "modern surroundings": "Are there modern surroundings in the image, such as skyscrapers,
          cars, or items that didn't exist in the 19th century? Answer with 'yes' or 'no'."
13   }
14 }
```

]

**Choice of LLM Backbone.**  To generate candidate anachronisms, we compared both proprietary and open-source LLMs of comparable scale: GPT-4o (proprietary), LLaMA-3.2–11B Instruct (open-source), Gemma-2–9B-it (open-source), and Claude 3.7 Sonnet (proprietary). As shown in Table 5, performance varies across coverage, redundancy, and runtime. Here, *coverage* ($|\mathcal{Z}|$) denotes the average number of candidate anachronisms proposed per prompt, while *redundancy* measures the proportion of these proposals that collapse into duplicates under fuzzy string matching (e.g., "cell phone," "mobile phone," and "smartphone" counted as one). GPT-4o achieves the best overall performance, with the highest coverage (8.9 anachronisms per prompt) and lowest redundancy (12.4%). LLaMA-3.2 performs nearly as well, while Gemma and Claude return fewer candidates with higher duplication, although Claude is the fastest overall. Given these results, we adopt GPT-4o as the default proposal backbone in our main pipeline.

## I  ANACHRONISM DETECTION

**Multi-VLM Detection.**  To avoid over-reliance on a single vision–language model, we reran the anachronism detection stage with three VLMs: GPT-4o, LLaMA-3.2–11B, and Qwen2.5 VL–7B. For each yes/no verification question, we collected predictions from all three models and selected

Table 5: Comparison of LLMs for anachronism proposal. $|\mathcal{Z}|$ = average number of proposed anachronisms per prompt; % Dup. = redundant lexical variants (fuzzy-matched at 0.8 similarity); TIME = average runtime per prompt (s). Best values in **bold**.

| Model | Version | $|\mathcal{Z}|$ | % Dup. | TIME |
|---|---|---|---|---|
| GPT | 4o (2024) | **8.9** | **12.4** | 7.6 |
| LLaMA | 3.2–11B Instruct | 8.5 | 13.1 | 8.2 |
| Gemma | 2–9B-it | 6.8 | 14.7 | 9.3 |
| Claude | 3.7 Sonnet | 7.2 | 14.9 | **6.4** |

the majority answer. This ensemble strategy increases robustness by reducing idiosyncratic errors of individual models. On our validation set with human annotations, the majority-vote approach achieved 75% agreement with human judgments, compared to 72% for GPT-4o alone, 68% for LLaMA-3.2, and 63% for Qwen2.5. All three models reliably identify clear anachronisms, such as smartphones, headphones, or modern appliances. Differences mainly occur in ambiguous cases, notably involving clothing styles or background elements lacking clear temporal indicators. Figure 12 illustrates representative examples of disagreement, partial agreement, and full consensus among the three VLMs.

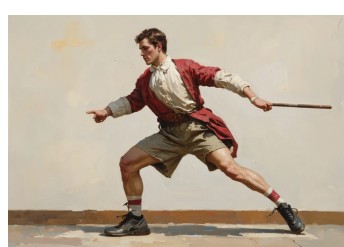 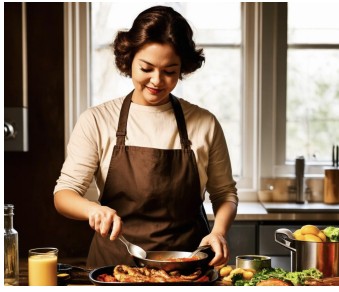 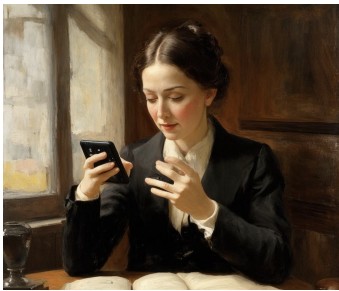

(a) Disagreement. Prompt: *"A person exercising in the 18th century."* GPT-4o and LLaMA-3.2 detect *modern clothing*, while Qwen2.5 answers *no*.

(b) Partial agreement. Prompt: *"A person cooking in the 18th century."* All three models detect *modern cooking equipment* (e.g., frying pan, glassware), but only GPT-4o and LLaMA-3.2 flag the entire *kitchen setting* as anachronistic, while Qwen2.5 does not.

(c) Full agreement. Prompt: *"A person communicating using a device in the 19th century."* GPT-4o, LLaMA-3.2, and Qwen2.5 all unanimously detect the presence of a *smartphone*, matching human judgment.

Figure 12: Representative examples of (a) disagreement, (b) partial agreement, and (c) full agreement among VLMs in the anachronism detection stage. Clear object-level anachronisms (e.g., *smartphones*) yield consistent consensus, while contextual or stylistic ambiguities (e.g., *clothing*, *kitchen setting*) lead to divergent responses across models.

## J  ANACHRONISM FREQUENCY & SEVERITY RESULTS

**Anachronism Frequency and Severity by Decade**  Figure 13 plots the top 15 anachronistic elements per model, ranked by *frequency* and *severity*, and aggregated across different decades rather than centuries. Each point corresponds to an element proposed by the LLM and verified via VQA, with fuzzy matching used to unify lexical variations. Items like *clothing* continue to appear more frequently than they do severely (7-10%), whereas elements such as *audio devices* are comparatively rare but exhibit higher severity, once generated, they strongly conflict with the historical context. This pattern again illustrates that the models rely heavily on conceptual cues (e.g., "audio sources" for music-related prompts), often at the expense of accurate chronological conditioning.

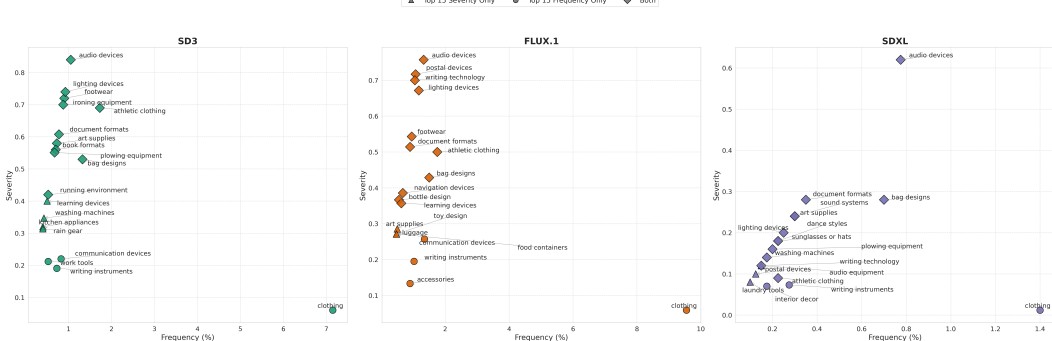

Figure 13: Top 15 anachronistic elements per model, aggregated across all decades, ranked by *frequency* (x-axis) and *severity* (y-axis). Circles indicate elements in the top 15 by frequency only, triangles by severity only, and diamonds appear in the top 15 of both.

**Overall Anachronism Frequency**    To assess each model's overall anachronistic tendency across historical periods, we measure the overall anachronism rate, defined as the proportion of images containing at least one detected anachronism. The frequency of specific anachronisms across centuries and decades for each model is illustrated in Figure 14. The 21st century is excluded as anachronisms cannot be meaningfully defined. By century, SD3's anachronism frequency increases from 15% in the 17th and 18th centuries to over 20% in the 19th century, then decreases to 17% in the 20th. FLUX.1 remains stable at 12–13%, while SDXL stays below 5%, with a slight uptick in the 20th century. By decade, SD3 peaks at 25% in the 1930s before falling to 10% in the 1990s. FLUX.1 follows a similar but milder trend, while SDXL remains consistently low (3.3–6.4%). These results suggest that while all models generate anachronisms, SD3 does so most frequently, especially in earlier periods. FLUX.1 remains stable, improving over time, while SDXL has the highest historical accuracy.

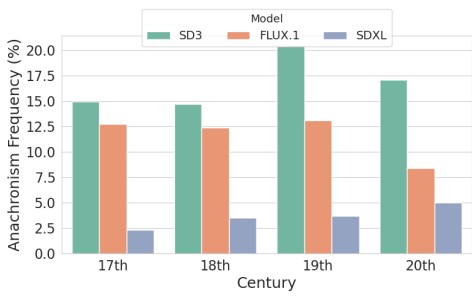

(a) Anachronism **frequency** across **centuries**. Each bar shows the percentage of images with at least one anachronism per model per century.

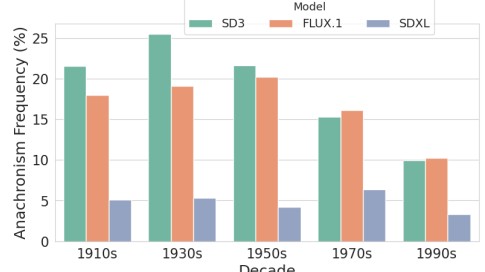

(b) Anachronism **frequency** across **decades**. Each bar shows the percentage of images with at least one anachronism per model per decade.

Figure 14: Anachronism frequency for each TTI model, grouped by centuries (left) and decades (right). Each model was evaluated on 1,000 images per period, totaling 4,000 per century and 5,000 per decade.

## J.1 ANACHRONISM SEVERITY AND FREQUENCY SCORES PER MODEL AND HISTORICAL PERIOD

| Anachronism | FLUX.1 Freq.(%) | Sev. | SD3 Freq.(%) | Sev. | SDXL Freq.(%) | Sev. |
|---|---|---|---|---|---|---|
| Clothing | 0.45 | 0.015 | 0.725 | 0.032 | 0.075 | 0.003 |
| Bag Designs | 0.325 | 0.65 | 0.475 | 1.0 | 0.075 | 0.15 |
| Audio Devices | 0.0 | 0.0 | 0.125 | 0.5 | 0.125 | 0.5 |
| Communication Devices | 0.225 | 0.3 | 0.1 | 0.133 | 0.0 | 0.0 |
| Ironing Equipment | 0.175 | 0.7 | 0.25 | 1.0 | 0.05 | 0.2 |
| Bottle Design | 0.25 | 1.0 | 0.175 | 0.7 | 0.025 | 0.1 |
| Food Containers | 0.15 | 0.6 | 0.2 | 0.8 | 0.0 | 0.0 |
| Lighting Devices | 0.1 | 0.2 | 0.075 | 0.3 | 0.05 | 0.2 |
| Vendor Setup | 0.15 | 0.6 | 0.225 | 0.9 | 0.0 | 0.0 |
| Art Supplies | 0.2 | 0.4 | 0.125 | 0.5 | 0.025 | 0.1 |
| Sound Equipment | 0.175 | 0.7 | 0.075 | 0.3 | 0.0 | 0.0 |
| Writing Instruments | 0.175 | 0.233 | 0.075 | 0.1 | 0.0 | 0.0 |
| Gallery Setting | 0.175 | 0.35 | 0.075 | 0.333 | 0.075 | 0.3 |
| Gift Wrapping Materials | 0.15 | 0.6 | 0.0 | 0.0 | 0.025 | 0.1 |
| Rain Gear | 0.175 | 0.35 | 0.15 | 0.6 | 0.05 | 0.2 |
| Ambient Lighting | 0.1 | 0.4 | 0.075 | 0.3 | 0.0 | 0.0 |
| Postal Devices | 0.075 | 0.3 | 0.075 | 0.3 | 0.0 | 0.0 |
| Athletic Clothing | 0.0 | 0.0 | 0.025 | 0.05 | 0.0 | 0.0 |
| Writing Technology | 0.075 | 0.3 | 0.075 | 0.3 | 0.0 | 0.0 |
| Pharmaceutical Packaging | 0.075 | 0.15 | 0.075 | 0.3 | 0.0 | 0.0 |
| Building Materials | 0.05 | 0.2 | 0.125 | 0.5 | 0.025 | 0.1 |
| Learning Devices | 0.0 | 0.0 | 0.0 | 0.0 | 0.0 | 0.0 |
| Maps or Interfaces | 0.0 | 0.0 | 0.0 | 0.0 | 0.0 | 0.0 |
| Navigating Devices | 0.0 | 0.05 | 0.0 | 0.0 | 0.0 | 0.0 |
| Running Environment | 0.0 | 0.0 | 0.0 | 0.0 | 0.0 | 0.0 |
| Street Fixtures | 0.075 | 0.3 | 0.0 | 0.0 | 0.0 | 0.0 |
| Toy Design | 0.05 | 0.2 | 0.05 | 0.2 | 0.0 | 0.0 |
| Watering Equipment | 0.075 | 0.15 | 0.05 | 0.2 | 0.0 | 0.0 |

Table 6: **17th Century** Anachronism Scores

| FLUX.1 Freq.(%) | Sev. | SD3 Freq.(%) | Sev. | SDXL Freq.(%) | Sev. |
|---|---|---|---|---|---|
| 0.350 | 0.015 | 0.625 | 0.026 | 0.075 | 0.003 |
| 0.325 | 0.65 | 0.425 | 0.85 | 0.100 | 0.20 |
| 0.000 | 0.00 | 0.175 | 0.70 | 0.225 | 1.0 |
| 0.200 | 0.267 | 0.175 | 0.233 | 0.000 | 0.00 |
| 0.025 | 0.10 | 0.225 | 1.0 | 0.025 | 0.10 |
| 0.200 | 0.80 | 0.050 | 0.20 | 0.050 | 0.20 |
| 0.150 | 0.60 | 0.200 | 0.80 | 0.000 | 0.00 |
| 0.100 | 0.40 | 0.125 | 0.50 | 0.050 | 0.20 |
| 0.100 | 0.40 | 0.100 | 0.40 | 0.000 | 0.00 |
| 0.075 | 0.30 | 0.050 | 0.20 | 0.100 | 0.40 |
| 0.200 | 0.80 | 0.075 | 0.30 | 0.000 | 0.00 |
| 0.075 | 0.10 | 0.125 | 0.50 | 0.000 | 0.00 |
| 0.100 | 0.10 | 0.075 | 0.30 | 0.025 | 0.10 |
| 0.050 | 0.20 | 0.175 | 0.70 | 0.025 | 0.10 |
| 0.100 | 0.40 | 0.100 | 0.40 | 0.000 | 0.00 |
| 0.100 | 0.40 | 0.125 | 0.50 | 0.000 | 0.00 |
| 0.200 | 0.80 | 0.050 | 0.20 | 0.000 | 0.00 |
| 0.050 | 0.10 | 0.250 | 0.50 | 0.000 | 0.00 |
| 0.200 | 0.80 | 0.000 | 0.00 | 0.000 | 0.00 |
| 0.000 | 0.00 | 0.125 | 0.50 | 0.000 | 0.00 |
| 0.025 | 0.10 | 0.125 | 0.50 | 0.050 | 0.20 |
| 0.000 | 0.00 | 0.100 | 0.40 | 0.000 | 0.00 |
| 0.000 | 0.00 | 0.025 | 0.025 | 0.000 | 0.00 |
| 0.000 | 0.00 | 0.025 | 0.10 | 0.000 | 0.00 |
| 0.000 | 0.00 | 0.050 | 0.20 | 0.000 | 0.00 |
| 0.025 | 0.10 | 0.025 | 0.10 | 0.050 | 0.20 |
| 0.000 | 0.00 | 0.075 | 0.30 | 0.025 | 0.10 |
| 0.025 | 0.10 | 0.025 | 0.10 | 0.025 | 0.10 |

Table 7: **18th Century** Anachronism Scores

| Anachronism | FLUX.1 Freq.(%) | Sev. | SD3 Freq.(%) | Sev. | SDXL Freq.(%) | Sev. |
|---|---|---|---|---|---|---|
| Clothing | 0.850 | 0.036 | 2.225 | 0.094 | 0.250 | 0.011 |
| Bag Designs | 0.275 | 0.55 | 0.400 | 0.80 | 0.125 | 0.00 |
| Audio Devices | 0.075 | 0.30 | 0.200 | 1.0 | 0.200 | 0.00 |
| Communication Devices | 0.225 | 0.30 | 0.175 | 0.233 | 0.000 | 0.00 |
| Ironing Equipment | 0.025 | 0.10 | 0.200 | 1.0 | 0.050 | 0.20 |
| Bottle Design | 0.175 | 0.70 | 0.050 | 0.20 | 0.025 | 0.10 |
| Food Containers | 0.150 | 0.60 | 0.100 | 0.40 | 0.000 | 0.00 |
| Lighting Devices | 0.100 | 0.40 | 0.125 | 0.50 | 0.050 | 0.20 |
| Vendor Setup | 0.100 | 0.40 | 0.100 | 0.40 | 0.000 | 0.00 |
| Art Supplies | 0.075 | 0.30 | 0.050 | 0.20 | 0.100 | 0.40 |
| Sound Equipment | 0.200 | 0.80 | 0.075 | 0.30 | 0.000 | 0.00 |
| Writing Instruments | 0.075 | 0.10 | 0.125 | 0.50 | 0.000 | 0.00 |
| Gallery Setting | 0.100 | 0.10 | 0.075 | 0.30 | 0.025 | 0.10 |
| Gift Wrapping Materials | 0.050 | 0.20 | 0.175 | 0.70 | 0.025 | 0.10 |
| Rain Gear | 0.100 | 0.40 | 0.100 | 0.40 | 0.000 | 0.00 |
| Ambient Lighting | 0.100 | 0.40 | 0.125 | 0.50 | 0.000 | 0.00 |
| Postal Devices | 0.200 | 0.80 | 0.050 | 0.20 | 0.000 | 0.00 |
| Athletic Clothing | 0.050 | 0.10 | 0.250 | 0.50 | 0.000 | 0.00 |
| Writing Technology | 0.200 | 0.80 | 0.000 | 0.00 | 0.000 | 0.00 |
| Pharmaceutical Packaging | 0.000 | 0.00 | 0.125 | 0.50 | 0.000 | 0.00 |
| Building Materials | 0.025 | 0.10 | 0.125 | 0.50 | 0.050 | 0.20 |
| Learning Devices | 0.000 | 0.00 | 0.100 | 0.40 | 0.000 | 0.00 |
| Maps or Interfaces | 0.000 | 0.00 | 0.025 | 0.025 | 0.000 | 0.00 |
| Navigation Devices | 0.000 | 0.00 | 0.025 | 0.10 | 0.000 | 0.00 |
| Running Environment | 0.000 | 0.00 | 0.050 | 0.20 | 0.000 | 0.00 |
| Street Fixtures | 0.025 | 0.10 | 0.025 | 0.10 | 0.050 | 0.20 |
| Toy Design | 0.000 | 0.00 | 0.075 | 0.30 | 0.025 | 0.10 |
| Watering Equipment | 0.025 | 0.10 | 0.025 | 0.10 | 0.025 | 0.10 |

Table 8: **19th Century** Anachronism Scores

| FLUX.1 Freq.(%) | Sev. | SD3 Freq.(%) | Sev. | SDXL Freq.(%) | Sev. |
|---|---|---|---|---|---|
| 0.400 | 0.017 | 0.725 | 0.031 | 0.125 | 0.005 |
| 0.225 | 0.45 | 0.300 | 0.60 | 0.175 | 0.35 |
| 0.225 | 0.90 | 0.225 | 0.90 | 0.150 | 0.60 |
| 0.150 | 0.20 | 0.100 | 0.133 | 0.050 | 0.067 |
| 0.075 | 0.30 | 0.175 | 0.70 | 0.050 | 0.20 |
| 0.075 | 0.30 | 0.100 | 0.40 | 0.000 | 0.00 |
| 0.075 | 0.30 | 0.100 | 0.40 | 0.000 | 0.00 |
| 0.150 | 0.60 | 0.200 | 0.80 | 0.000 | 0.00 |
| 0.075 | 0.30 | 0.100 | 0.40 | 0.000 | 0.00 |
| 0.025 | 0.10 | 0.000 | 0.00 | 0.000 | 0.00 |
| 0.025 | 0.10 | 0.050 | 0.222 | 0.000 | 0.00 |
| 0.025 | 0.033 | 0.125 | 0.167 | 0.000 | 0.00 |
| 0.025 | 0.10 | 0.050 | 0.20 | 0.075 | 0.30 |
| 0.000 | 0.00 | 0.000 | 0.00 | 0.000 | 0.00 |
| 0.000 | 0.00 | 0.000 | 0.00 | 0.000 | 0.00 |
| 0.000 | 0.00 | 0.000 | 0.50 | 0.000 | 0.00 |
| 0.025 | 0.10 | 0.000 | 0.00 | 0.025 | 0.10 |
| 0.025 | 0.05 | 0.225 | 0.45 | 0.000 | 0.00 |
| 0.025 | 0.10 | 0.000 | 0.00 | 0.025 | 0.10 |
| 0.075 | 0.30 | 0.050 | 0.20 | 0.000 | 0.00 |
| 0.000 | 0.00 | 0.000 | 0.00 | 0.000 | 0.00 |
| 0.050 | 0.20 | 0.225 | 0.90 | 0.075 | 0.30 |
| 0.050 | 0.20 | 0.250 | 1.00 | 0.125 | 0.50 |
| 0.075 | 0.30 | 0.250 | 1.00 | 0.125 | 0.50 |
| 0.075 | 0.30 | 0.100 | 0.40 | 0.025 | 0.10 |
| 0.025 | 0.10 | 0.025 | 0.10 | 0.000 | 0.00 |
| 0.075 | 0.30 | 0.025 | 0.10 | 0.000 | 0.00 |
| 0.000 | 0.00 | 0.000 | 0.00 | 0.000 | 0.00 |

Table 9: **20th Century** Anachronism Scores

# K    EXAMPLE OUTPUTS OF SEVERE ANACHRONISMS

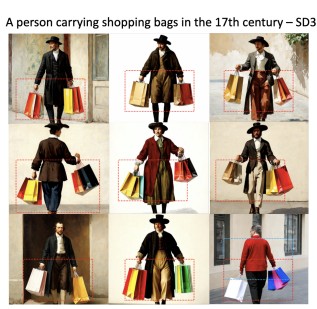

(a) Modern shopping bags

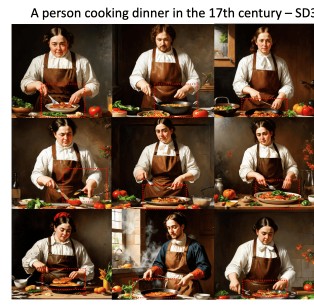

(b) Modern cooking tool

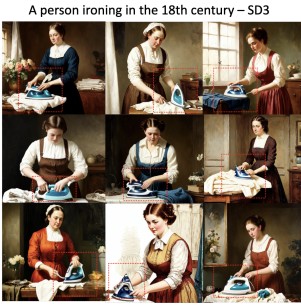

(c) Modern ironing equipment

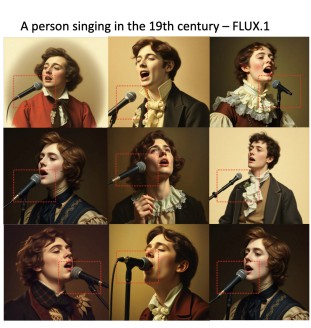

(d) Modern audio device

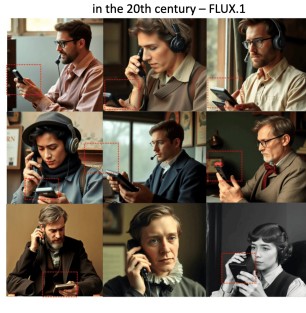

(e) Modern communication device

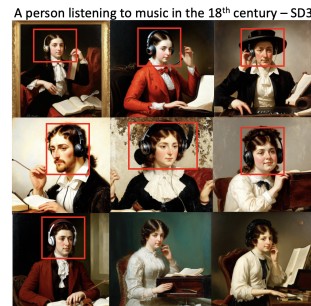

(f) Modern audio device

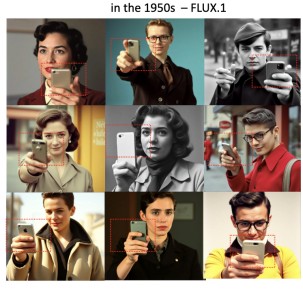

(g) Modern communication device

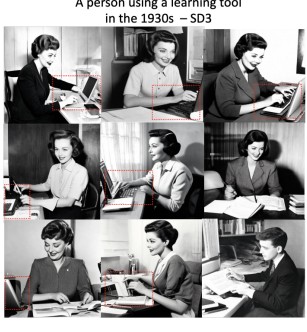

(h) Modern learning tool

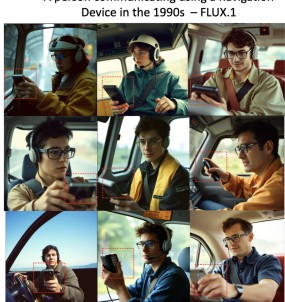

(i) Modern navigation device

Figure 15: Examples of generated images depicting universal human activities across different historical periods, highlighting instances of high-severity anachronisms.

# L  USER STUDY

Figure 16 presents the user interface for the human evaluation study, which includes the task description given to participants and example questions. The task description introduces the concept of anachronisms to ensure participants understand it before beginning the task and provides illustrative examples of both historically accurate and anachronistic images for further clarification. Participants were then required to assess TTI-generated images for chronological inconsistencies by answering a series of yes-no questions.

Figure 16: User interface for the human evaluation study, presenting the task description given to participants along with example questions. The interface explains anachronisms, provides illustrative examples of historically accurate and anachronistic images, and includes yes-no questions for assessing chronological inconsistencies in TTI-generated images.

## M    LLM INPUT FOR DEMOGRAPHIC PREDICTION

For a given activity and historical period described in the prompt, estimate the plausible demographic breakdown based on global historical context:

- **Gender:(%)** Proportion of male and female participants
- **Race:(%)** Estimated distribution across the following categories: White, Black, Asian, Indian, East Asian, Southeast Asian, and Middle Eastern.

The estimates should be informed by historical social structures, legal constraints, cultural practices, and prevailing social hierarchies. Avoid focusing on a specific geographic region (e.g., Europe); instead, consider patterns and norms that shaped access and participation globally across different societies.

## N    FAIRFACE-DERIVED DEMOGRAPHIC OUTPUTS

We analyze the face classifier predictions from the three TTI models $\mathbf{m} \in \{\text{SDXL}, \text{SD3}, \text{FLUX.1}\}$ to understand the distributions they produce before comparing them to GPT-4o's estimates. Out of 30,000 generated images, 1,221 (4.07%) were excluded due to low classifier confidence, often because the faces were blurred or unclear.

**Comparison to DeepFace's Predictions**    To assess FairFace's consistency, we sampled 5,000 images (500 per time period/model pair) and found that FairFace and DeepFace agreed on gender 95.9% of the time ($\kappa = 0.93$) and on race 90.8% of the time ($\kappa = 0.83$). To align FairFace's seven race categories with DeepFace's single 'Asian' label, we aggregated FairFace's 'East Asian' and 'Southeast Asian' predictions into one 'Asian' category. All other FairFace labels map directly. After this, we computed the per-class agreement rates and Cohen's $\kappa$ between the two face classifiers reported in Table 10.

Table 10:    Per-class agreement rates and Cohen's $\kappa$ between FairFace and DeepFace (post-aggregation).

| Category | Percent agreement (%) | Cohen's $\kappa$ |
|---|---|---|
| White | 93.0 | 0.85 |
| Black | 94.5 | 0.87 |
| Indian | 87.2 | 0.75 |
| Asian | 92.0 | 0.88 |
| Middle Eastern | 88.8 | 0.80 |
| Latino | 89.0 | 0.81 |
| Male | 95.6 | 0.92 |
| Female | 96.2 | 0.93 |

**Face Classifier Predictions**    Examining these classifier-derived outputs reveals that demographic patterns vary across models. As shown in the barplots below, SDXL primarily generates male figures in earlier centuries (e.g., 70 women versus 930 men in the 17th century), while SD3, in contrast, produces more female figures in certain decades, such as the 1950s (700 female versus 300 male). Across all models, white individuals are overwhelmingly depicted in earlier periods, with racial diversity increasing only after the 20th century, indicating potential biases in how different models represent historical demographics.

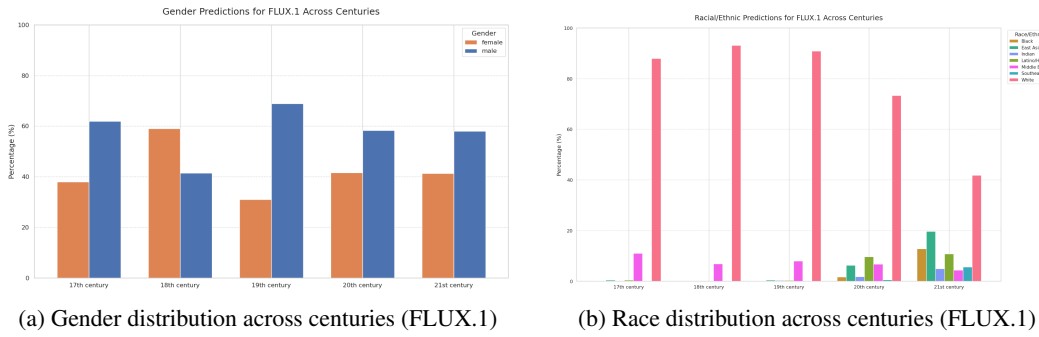

(a) Gender distribution across centuries (FLUX.1)   (b) Race distribution across centuries (FLUX.1)

Figure 17: Demographic breakdowns (FairFace predictions) across centuries for **FLUX.1**.

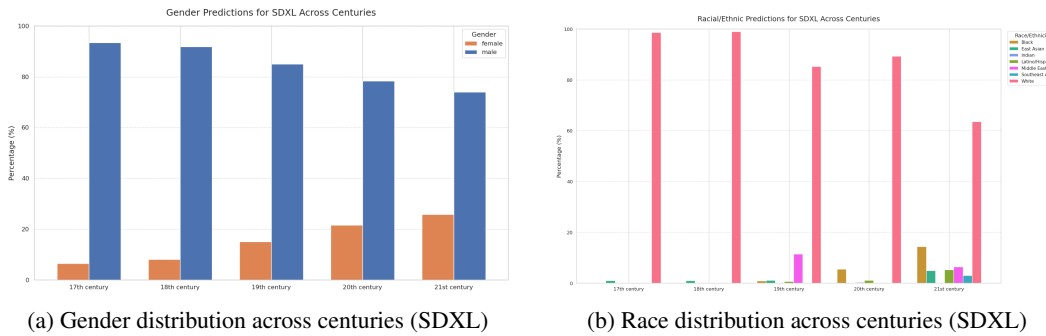

(a) Gender distribution across centuries (SDXL)   (b) Race distribution across centuries (SDXL)

Figure 18: Demographic breakdowns (FairFace predictions) across centuries for **SDXL**.

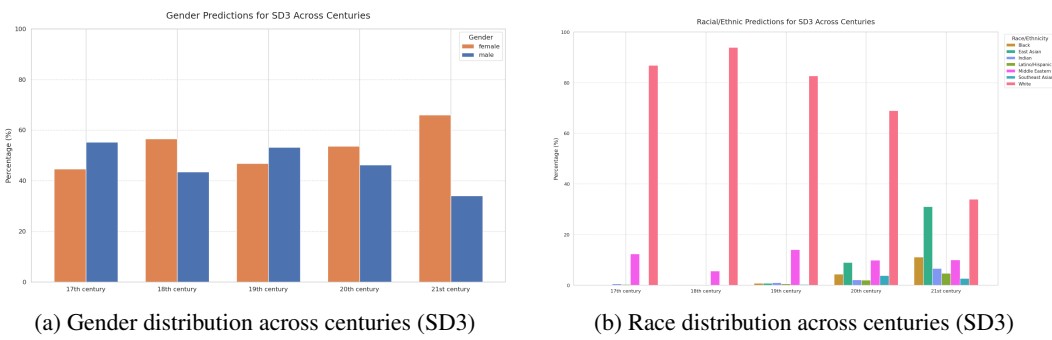

(a) Gender distribution across centuries (SD3)   (b) Race distribution across centuries (SD3)

Figure 19: Demographic breakdowns (FairFace predictions) across centuries for **SD3**.

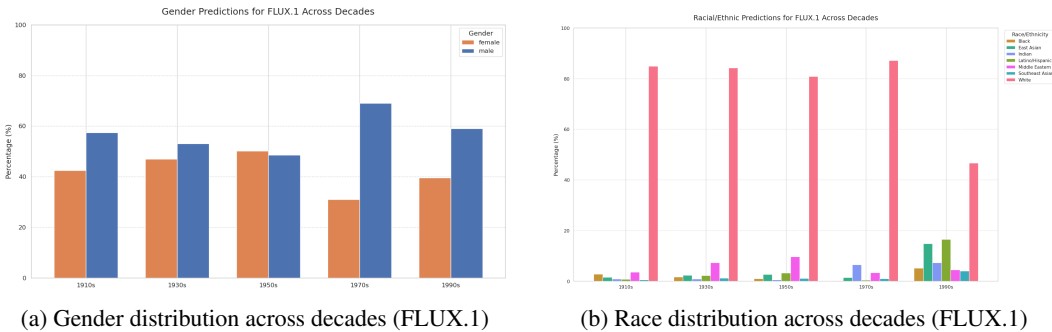

(a) Gender distribution across decades (FLUX.1)   (b) Race distribution across decades (FLUX.1)

Figure 20: Demographic breakdowns (FairFace predictions) across decades for **FLUX.1**.

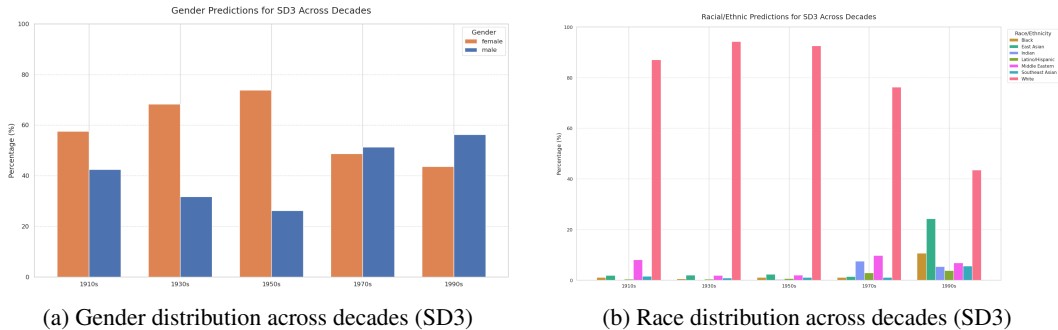

(a) Gender distribution across decades (SD3)   (b) Race distribution across decades (SD3)

Figure 21: Demographic breakdowns (FairFace predictions) across decades for **SD3**.

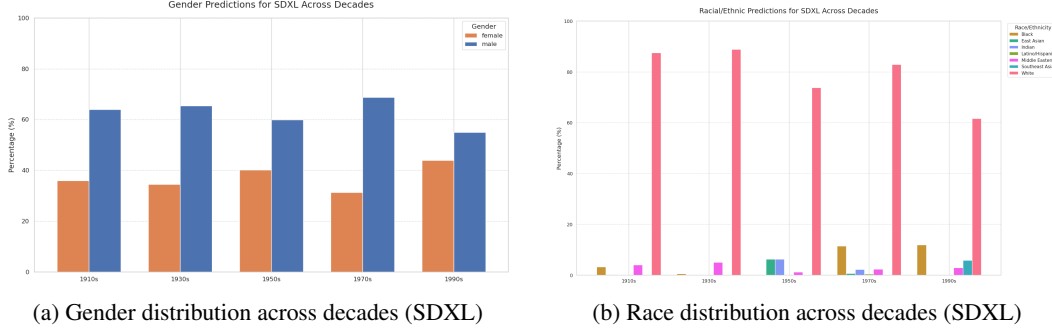

(a) Gender distribution across decades (SDXL)   (b) Race distribution across decades (SDXL)

Figure 22: Demographic breakdowns (FairFace predictions) across decades for **SDXL**.

## O  VALIDATION OF DEMOGRAPHIC ESTIMATION WITH HISTORICAL DATA

To further validate the credibility of the LLMs used for demographic estimation, we conducted an additional evaluation comparing model predictions with historical statistics from Our World in Data (OWID), a public platform compiling long-term global development indicators. Specifically, we examined model estimates for three universal activity categories in *HistVis*, Education, Agriculture, and Work & Collaboration, for which historical demographic data is available. While OWID provides limited coverage for the 17th–18th centuries, representative gender- and continent-level participation statistics are available from the 19th century onward.

Unlike our main pipeline, which prompts an LLM to provide racial demographic distributions for comparison with FairFace classifications, here we deliberately avoided mapping country-level OWID data to racial categories, as such mappings would be reductive. Instead, we used continent-level shares (Africa, North America, South America, Asia, Europe, Oceania), which offer a more robust and interpretable basis for comparison. For each prompt, four LLMs were asked to estimate demographic breakdowns by gender and continent; their responses were aggregated and compared against OWID data using Mean Absolute Error (MAE).

For this validation step, each LLM was prompted as follows:

> *For a given activity and historical period, estimate the plausible demographic breakdown based on global historical context:*
>
> - **Gender (%):** Proportion of male and female participants.
> - **Continent (%):** Estimated distribution across the following regions: Africa, Asia, Europe, North America, South America, Oceania.
>
> *Base your estimates on historical population distributions, social structures, and participation norms relevant to the activity, and avoid assuming a single geographic focus.*

Overall results show that GPT-4o achieved the lowest MAE (4.64), followed closely by LLaMA-3.2 (4.83), then Claude 3.7 (6.93), and Qwen2.5 VL-7B (7.66). This indicates that while there is variation across models, all produce estimates broadly consistent with historical distributions. These results, not included in the original submission, strengthen the robustness of our demographic estimation pipeline and justify our selection of GPT-4o as the baseline. While LLM outputs are not substitutes for expert historical analysis, they provide plausible proxies for detecting extreme over- or under-representation patterns (e.g., overrepresentation of men in Cooking during early centuries, or delayed inclusion of Asian and Middle Eastern populations in Religion and Agriculture until the 21st century).

Below we report aggregated MAE scores across gender and continent-level distributions as well as disaggregated by time period for each model.

Table 11: **Aggregated MAE** scores across gender and continent-level distributions.

| Model | Male | Female | Europe | Africa | N.Am. | S.Am. | Asia | Oceania | MAE |
|-------|------|--------|--------|--------|-------|-------|------|---------|-----|
| LLaMA-3.2 | 2.49 | 2.49 | 8.72 | 3.54 | 5.96 | 4.31 | 6.09 | 5.02 | 4.83 |
| Qwen2.5 VL-7B | 5.02 | 5.02 | 13.80 | 5.14 | 9.71 | 7.56 | 6.51 | 8.54 | 7.66 |
| Claude 3.7 | 2.89 | 2.89 | 16.54 | 3.04 | 3.96 | 6.34 | 12.79 | 6.96 | 6.93 |
| GPT-4o | 1.90 | 1.90 | 8.75 | 3.05 | 6.47 | 4.58 | 4.66 | 5.80 | 4.64 |

Table 12: Time-period MAE scores for **GPT-4o**.

| Period | Male | Female | Europe | Africa | N.Am. | Asia | S.Am. | Oceania |
|--------|------|--------|--------|--------|-------|------|-------|---------|
| 1910s | 2.85 | 2.85 | 3.83 | 0.64 | 4.17 | 8.46 | 3.20 | 4.80 |
| 1930s | 0.92 | 0.92 | 4.23 | 1.09 | 7.95 | 3.81 | 4.90 | 6.10 |
| 1950s | 3.74 | 3.74 | 5.90 | 1.50 | 6.45 | 2.04 | 4.80 | 5.90 |
| 1970s | 2.86 | 2.86 | 9.24 | 3.16 | 5.09 | 0.49 | 4.30 | 5.70 |
| 1990s | 1.01 | 1.01 | 12.34 | 6.58 | 8.00 | 1.74 | 5.10 | 6.20 |
| 19th c. | 2.56 | 2.56 | 17.13 | 2.01 | 3.52 | 12.50 | 6.40 | 7.80 |
| 20th c. | 1.27 | 1.27 | 8.14 | 3.36 | 7.10 | 1.92 | 4.10 | 5.30 |
| 21st c. | 0.00 | 0.00 | 9.22 | 6.09 | 9.47 | 6.33 | 3.80 | 4.60 |

Table 13: Time-period MAE scores for **LLaMA 3.2–11B**.

| Period | Male | Female | Europe | Africa | N.Am. | Asia | S.Am. | Oceania |
|--------|------|--------|--------|--------|-------|------|-------|---------|
| 1910s | 2.65 | 2.65 | 16.33 | 8.54 | 0.67 | 8.46 | 2.10 | 3.20 |
| 1930s | 4.58 | 4.58 | 8.27 | 4.91 | 5.45 | 8.81 | 3.80 | 4.60 |
| 1950s | 2.26 | 2.26 | 1.60 | 2.50 | 5.45 | 4.54 | 4.90 | 5.80 |
| 1970s | 1.14 | 1.14 | 1.74 | 3.34 | 6.09 | 1.01 | 5.20 | 6.10 |
| 1990s | 1.99 | 1.99 | 4.84 | 0.08 | 9.00 | 4.24 | 4.30 | 4.70 |
| 19th c. | 0.06 | 0.06 | 27.63 | 6.31 | 6.92 | 14.40 | 6.80 | 7.30 |
| 20th c. | 6.27 | 6.27 | 2.64 | 0.04 | 6.10 | 3.42 | 3.50 | 4.10 |
| 21st c. | 1.00 | 1.00 | 6.72 | 2.59 | 7.97 | 3.83 | 3.90 | 4.40 |

Table 14: Time-period MAE scores for **Qwen2.5 VL-7B**.

| Period | Male | Female | Europe | Africa | N.Am. | Asia | S.Am. | Oceania |
|--------|------|--------|--------|--------|-------|------|-------|---------|
| 1910s | 6.35 | 6.35 | 3.67 | 4.54 | 5.67 | 2.54 | 6.20 | 7.10 |
| 1930s | 4.42 | 4.42 | 11.73 | 5.91 | 10.45 | 7.19 | 8.10 | 9.20 |
| 1950s | 7.74 | 7.74 | 13.40 | 5.50 | 10.45 | 8.46 | 7.80 | 8.90 |
| 1970s | 6.86 | 6.86 | 12.74 | 7.34 | 11.09 | 8.99 | 6.90 | 7.80 |
| 1990s | 3.01 | 3.01 | 15.84 | 3.92 | 14.00 | 5.76 | 8.30 | 9.10 |
| 19th c. | 10.06 | 10.06 | 13.63 | 8.31 | 1.92 | 3.40 | 9.40 | 10.60 |
| 20th c. | 0.73 | 0.73 | 17.64 | 3.04 | 11.10 | 9.58 | 7.00 | 8.20 |
| 21st c. | 1.00 | 1.00 | 21.72 | 2.59 | 12.97 | 6.17 | 6.80 | 7.40 |

Table 15: Time-period MAE scores for **Claude 3.7**.

| Period | Male | Female | Europe | Africa | N.Am. | Asia | S.Am. | Oceania |
|--------|------|--------|--------|--------|-------|------|-------|---------|
| 1910s | 5.55 | 5.55 | 28.33 | 5.54 | 5.73 | 17.06 | 4.80 | 5.90 |
| 1930s | 0.42 | 0.42 | 18.27 | 3.91 | 0.45 | 14.81 | 6.10 | 7.40 |
| 1950s | 1.34 | 1.34 | 13.60 | 3.50 | 2.45 | 12.54 | 6.80 | 7.20 |
| 1970s | 1.06 | 1.06 | 8.26 | 1.34 | 3.09 | 10.01 | 6.50 | 6.80 |
| 1990s | 0.41 | 0.41 | 3.16 | 1.08 | 1.00 | 5.24 | 6.20 | 6.90 |
| 19th c. | 14.06 | 14.06 | 46.63 | 6.31 | 14.92 | 25.40 | 8.70 | 9.10 |
| 20th c. | 0.27 | 0.27 | 12.36 | 0.04 | 1.10 | 13.42 | 5.90 | 6.30 |
| 21st c. | 0.00 | 0.00 | 1.72 | 2.59 | 2.97 | 3.83 | 5.70 | 6.10 |

# P   COMPARATIVE RESULTS WITH LLaMA 3.2–11B

Tables 16 and 17 report average demographic deviations across activity categories for **FLUX.1 Schnell**, using **GPT-4o** and **LLaMA 3.2–11B** as demographic estimators. Deviations are computed by comparing FairFace classifier predictions on generated images against LLM-derived historical baselines. Each entry shows the degree of under- (U) or over-representation (O) for a given group, with the rightmost column (**Avg**) summarizing the mean absolute deviation across groups. Both estimators reveal consistent imbalances: categories such as *art*, *agriculture*, and *craftsmanship & labor* show strong over-representation of male and White figures, while domains such as *family & relationships* or *communication & social interaction* exhibit more moderate deviations. Crucially, the overall trends remain qualitatively similar across GPT-4o and LLaMA-3.2, underscoring LLaMA-3.2 as a viable open-source alternative.

Table 16: Average demographic deviations per category using **GPT-4o** as demographic estimator. Values show under- and over-representation for each demographic group, plus overall average deviation.

| Category | U-M | U-F | U-W | U-B | U-Lat | U-EAs | U-SEAs | U-Ind | U-ME | O-M | O-F | O-W | O-B | O-Lat | O-EAs | O-SEAs | O-Ind | O-ME | Avg |
|---|---|---|---|---|---|---|---|---|---|---|---|---|---|---|---|---|---|---|---|
| Agriculture | 18.13 | 0.74 | 0.0 | 13.89 | 1.86 | 28.95 | 11.29 | 19.86 | 0.6 | 0.74 | 17.93 | 57.97 | 5.04 | 0.1 | 0.0 | 0.0 | 0.0 | 13.08 | **10.57** |
| Art | 43.57 | 0.0 | 0.0 | 12.6 | 2.77 | 23.30 | 8.8 | 21.57 | 2.0 | 0.0 | 43.57 | 68.07 | 0.0 | 0.43 | 0.0 | 0.0 | 0.0 | 2.54 | **12.73** |
| Celebration & Festivities | 17.72 | 0.0 | 0.0 | 11.17 | 3.86 | 22.61 | 8.06 | 22.26 | 0.19 | 0.0 | 17.72 | 60.12 | 0.0 | 0.0 | 0.0 | 0.0 | 0.0 | 8.04 | **9.54** |
| Commerce | 19.74 | 2.0 | 2.89 | 4.49 | 3.75 | 12.89 | 3.2 | 13.24 | 2.6 | 2.0 | 19.74 | 35.22 | 0.11 | 0.05 | 3.42 | 1.56 | 0.0 | 2.71 | **7.20** |
| Communication & Social Interaction | 3.2 | 7.66 | 2.0 | 4.97 | 5.0 | 14.0 | 5.4 | 14.29 | 1.4 | 9.46 | 3.2 | 32.15 | 1.16 | 0.0 | 1.32 | 0.0 | 0.0 | 12.45 | **6.54** |
| Cooking & Dining | 5.34 | 19.34 | 0.4 | 9.6 | 2.4 | 18.82 | 7.35 | 22.2 | 3.4 | 19.34 | 5.34 | 58.68 | 0.2 | 1.49 | 3.8 | 0.0 | 0.0 | 0.0 | **9.87** |
| Craftsmanship & Labor (set 2) | 28.69 | 2.0 | 1.6 | 20.42 | 2.67 | 24.0 | 12.4 | 19.22 | 1.87 | 0.0 | 10.69 | 51.10 | 0.0 | 0.0 | 4.0 | 0.0 | 0.0 | 7.08 | **10.32** |
| Daily Chores & Activities | 14.29 | 7.33 | 0.0 | 12.6 | 3.7 | 19.0 | 8.8 | 22.2 | 1.18 | 7.33 | 14.29 | 63.32 | 0.0 | 0.02 | 3.33 | 0.0 | 0.0 | 0.81 | **9.90** |
| Education & Learning | 31.91 | 0.0 | 0.0 | 4.93 | 5.08 | 9.0 | 4.8 | 14.66 | 2.65 | 0.0 | 31.91 | 34.45 | 0.96 | 0.0 | 5.70 | 0.0 | 0.0 | 0.0 | **8.11** |
| Emotion & Personal Traits | 5.39 | 10.89 | 0.0 | 9.0 | 2.01 | 20.11 | 8.8 | 22.17 | 1.89 | 10.89 | 5.39 | 60.14 | 0.42 | 3.17 | 0.83 | 0.0 | 0.0 | 0.11 | **8.96** |
| Family & Relationships | 14.20 | 0.0 | 0.0 | 11.65 | 3.33 | 18.11 | 8.25 | 23.0 | 2.74 | 0.0 | 14.20 | 61.71 | 0.0 | 0.94 | 4.03 | 0.0 | 0.0 | 0.4 | **9.03** |
| Fashion & Personal Care | 11.20 | 19.0 | 0.0 | 12.6 | 2.37 | 21.4 | 8.8 | 21.10 | 1.81 | 19.0 | 11.20 | 63.97 | 0.0 | 0.0 | 2.62 | 0.0 | 0.0 | 1.48 | **10.92** |
| Health & Well-being | 34.82 | 2.0 | 0.0 | 7.6 | 4.4 | 12.86 | 4.8 | 15.4 | 3.77 | 2.0 | 34.82 | 41.71 | 0.0 | 1.86 | 0.0 | 0.0 | 0.0 | 5.27 | **9.52** |
| Music | 17.31 | 7.27 | 0.0 | 8.8 | 4.6 | 20.92 | 8.8 | 19.30 | 4.2 | 7.27 | 17.31 | 63.69 | 2.93 | 0.0 | 0.0 | 0.0 | 0.0 | 0.0 | **10.13** |
| Physical & Recreational Activities | 26.28 | 4.81 | 0.0 | 11.63 | 4.6 | 20.43 | 8.8 | 23.0 | 0.0 | 4.81 | 26.28 | 43.72 | 0.0 | 0.0 | 3.18 | 0.0 | 0.0 | 21.56 | **11.06** |
| Religion & Spirituality | 10.19 | 7.03 | 6.25 | 8.02 | 2.4 | 17.87 | 7.2 | 11.07 | 0.0 | 7.03 | 10.19 | 24.37 | 0.0 | 2.38 | 1.67 | 0.75 | 3.38 | 20.27 | **7.78** |
| Survival & Comfort | 25.40 | 0.0 | 0.0 | 7.2 | 2.49 | 17.84 | 7.69 | 21.89 | 2.6 | 0.0 | 25.40 | 51.55 | 4.93 | 0.18 | 0.88 | 0.0 | 0.0 | 2.18 | **9.46** |
| Transportation & Travel | 23.13 | 2.2 | 1.25 | 7.6 | 5.4 | 11.69 | 2.55 | 14.15 | 1.4 | 2.2 | 23.13 | 29.94 | 0.0 | 0.0 | 5.0 | 2.85 | 0.0 | 6.26 | **7.71** |
| Urban Life | 16.5 | 11.31 | 4.5 | 7.6 | 5.4 | 12.17 | 2.2 | 14.49 | 1.4 | 11.31 | 16.5 | 18.70 | 0.0 | 0.0 | 0.0 | 8.61 | 0.0 | 20.45 | **8.40** |
| Work & Collaboration | 16.49 | 3.42 | 2.5 | 5.6 | 5.4 | 12.13 | 4.8 | 14.15 | 1.98 | 3.42 | 16.49 | 30.01 | 5.5 | 0.0 | 0.0 | 0.0 | 0.0 | 11.04 | **7.38** |

Table 17: Average demographic deviations per category using **LLaMA 3.2–11B** as the demographic estimator. Values show under- and over-representation for each group, plus overall average deviation.

| Category | U-M | U-F | U-W | U-B | U-Lat | U-EAs | U-SEAs | U-Ind | U-ME | O-M | O-F | O-W | O-B | O-Lat | O-EAs | O-SEAs | O-Ind | O-ME | Avg |
|---|---|---|---|---|---|---|---|---|---|---|---|---|---|---|---|---|---|---|---|
| Agriculture | 17.6 | 0.9 | 0.0 | 14.2 | 2.1 | 29.4 | 11.6 | 20.2 | 0.7 | 0.9 | 18.4 | 58.5 | 5.2 | 0.0 | 0.0 | 0.0 | 0.0 | 12.7 | **10.7** |
| Art | 42.9 | 0.0 | 0.0 | 12.9 | 3.1 | 22.7 | 9.1 | 21.9 | 2.1 | 0.0 | 44.2 | 67.3 | 0.0 | 0.6 | 0.0 | 0.0 | 0.0 | 2.8 | **12.6** |
| Celebration & Festivities | 18.2 | 0.0 | 0.0 | 10.9 | 3.6 | 22.1 | 8.3 | 21.7 | 0.3 | 0.0 | 18.5 | 61.0 | 0.0 | 0.0 | 0.0 | 0.0 | 0.0 | 7.9 | **9.6** |
| Commerce | 19.2 | 2.1 | 3.0 | 4.7 | 4.0 | 13.3 | 3.5 | 12.7 | 2.8 | 2.1 | 20.1 | 34.8 | 0.2 | 0.0 | 3.2 | 1.8 | 0.0 | 2.6 | **7.4** |
| Communication & Social Interaction | 3.3 | 7.4 | 2.1 | 5.2 | 4.8 | 13.7 | 5.7 | 13.9 | 1.6 | 9.2 | 3.5 | 31.9 | 1.0 | 0.0 | 1.6 | 0.0 | 0.0 | 12.8 | **6.6** |
| Cooking & Dining | 5.6 | 18.9 | 0.6 | 9.9 | 2.5 | 19.3 | 7.7 | 21.8 | 3.2 | 18.8 | 5.1 | 59.2 | 0.1 | 1.3 | 3.9 | 0.0 | 0.0 | 0.0 | **9.9** |
| Craftsmanship & Labor (set 2) | 28.4 | 2.2 | 1.8 | 20.9 | 2.9 | 23.6 | 12.1 | 18.8 | 2.0 | 0.0 | 10.5 | 51.6 | 0.0 | 0.0 | 4.3 | 0.0 | 0.0 | 7.4 | **10.4** |
| Daily Chores & Activities | 14.0 | 7.6 | 0.0 | 12.9 | 3.8 | 19.3 | 9.1 | 21.9 | 1.2 | 7.6 | 14.1 | 63.6 | 0.0 | 0.0 | 3.2 | 0.0 | 0.0 | 0.7 | **9.8** |
| Education & Learning | 32.2 | 0.0 | 0.0 | 5.1 | 5.2 | 8.7 | 4.6 | 14.9 | 2.8 | 0.0 | 32.4 | 34.0 | 1.1 | 0.0 | 5.4 | 0.0 | 0.0 | 0.0 | **8.2** |
| Emotion & Personal Traits | 5.5 | 11.2 | 0.0 | 9.3 | 2.2 | 19.9 | 8.6 | 21.9 | 2.0 | 11.0 | 5.2 | 59.7 | 0.5 | 3.0 | 0.9 | 0.0 | 0.0 | 0.2 | **9.0** |
| Family & Relationships | 13.9 | 0.0 | 0.0 | 11.3 | 3.6 | 18.5 | 8.5 | 22.6 | 2.6 | 0.0 | 14.5 | 62.2 | 0.0 | 0.8 | 3.9 | 0.0 | 0.0 | 0.5 | **9.2** |
| Fashion & Personal Care | 11.4 | 18.6 | 0.0 | 12.3 | 2.6 | 21.7 | 9.0 | 21.2 | 2.0 | 19.2 | 11.0 | 64.4 | 0.0 | 0.0 | 2.4 | 0.0 | 0.0 | 1.7 | **10.8** |
| Health & Well-being | 34.1 | 2.2 | 0.0 | 7.9 | 4.2 | 13.0 | 5.0 | 15.1 | 3.9 | 2.2 | 34.5 | 41.9 | 0.0 | 2.0 | 0.0 | 0.0 | 0.0 | 5.1 | **9.6** |
| Music | 17.6 | 7.0 | 0.0 | 9.1 | 4.4 | 21.1 | 9.2 | 18.9 | 4.3 | 7.0 | 17.6 | 63.2 | 2.7 | 0.0 | 0.0 | 0.0 | 0.0 | 0.0 | **10.2** |
| Physical & Recreational Activities | 25.9 | 4.9 | 0.0 | 11.3 | 4.7 | 20.6 | 9.1 | 22.7 | 0.0 | 5.0 | 26.1 | 44.1 | 0.0 | 0.0 | 3.3 | 0.0 | 0.0 | 21.2 | **11.0** |
| Religion & Spirituality | 9.9 | 7.2 | 6.0 | 8.3 | 2.6 | 18.1 | 7.5 | 11.3 | 0.0 | 7.2 | 10.0 | 24.1 | 0.0 | 2.5 | 1.8 | 0.9 | 3.4 | 20.6 | **7.8** |
| Survival & Comfort | 25.0 | 0.0 | 0.0 | 7.4 | 2.6 | 17.6 | 7.9 | 21.7 | 2.7 | 0.0 | 25.2 | 51.0 | 5.2 | 0.0 | 0.9 | 0.0 | 0.0 | 2.0 | **9.5** |
| Transportation & Travel | 23.4 | 2.1 | 1.3 | 7.4 | 5.6 | 11.9 | 2.7 | 14.4 | 1.6 | 2.1 | 23.4 | 29.7 | 0.0 | 0.0 | 5.2 | 2.9 | 0.0 | 6.5 | **7.8** |
| Urban Life | 16.2 | 11.6 | 4.4 | 7.8 | 5.6 | 11.9 | 2.4 | 14.7 | 1.5 | 11.6 | 16.3 | 19.0 | 0.0 | 0.0 | 0.0 | 8.8 | 0.0 | 20.7 | **8.4** |
| Work & Collaboration | 16.7 | 3.6 | 2.6 | 5.8 | 5.7 | 11.9 | 5.0 | 13.9 | 2.0 | 3.6 | 16.8 | 29.7 | 5.7 | 0.0 | 0.0 | 0.0 | 0.0 | 11.3 | **7.4** |

# Q  GENDER AND RACIAL OVER-UNDERREPRESENTATION VALUES WITH GPT-4O

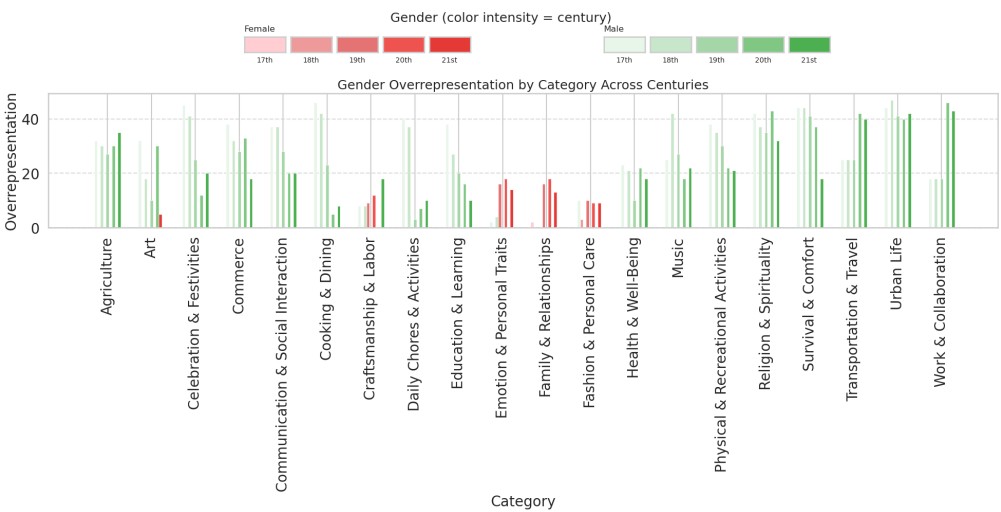

Figure 23: Over- and underrepresentation of gender predictions across **centuries** for **SDXL**.

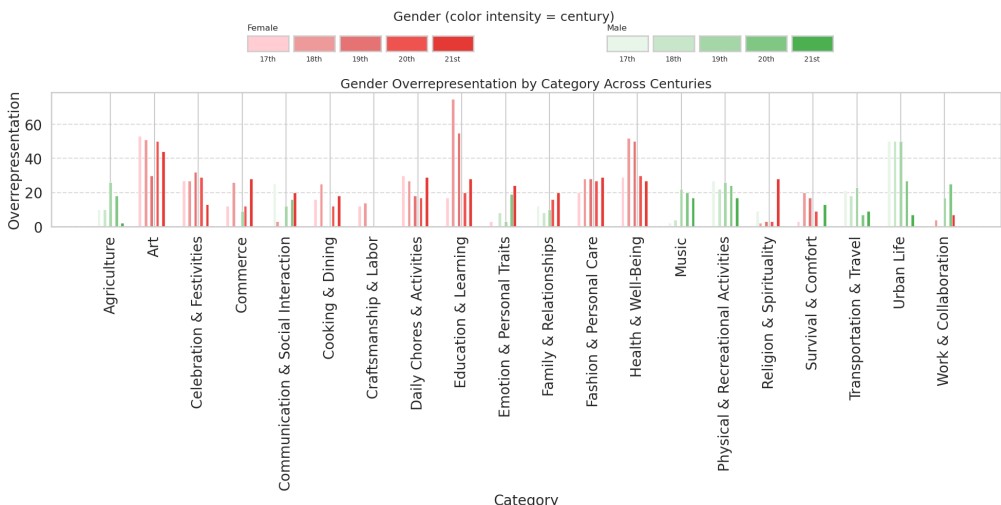

Figure 24: Over- and underrepresentation of gender predictions across **centuries** for **SD3**.

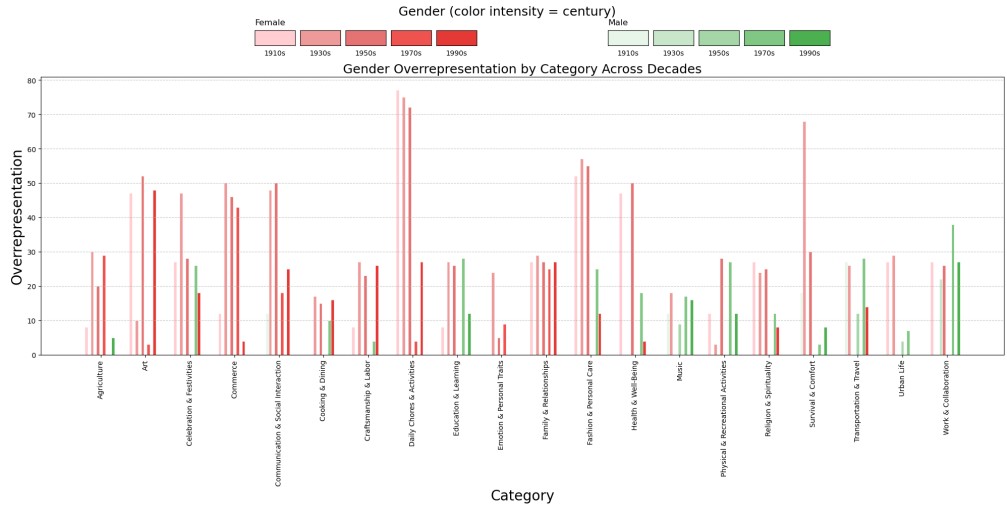

Figure 25: Over- and underrepresentation of gender predictions across **decades** for **SD3**.

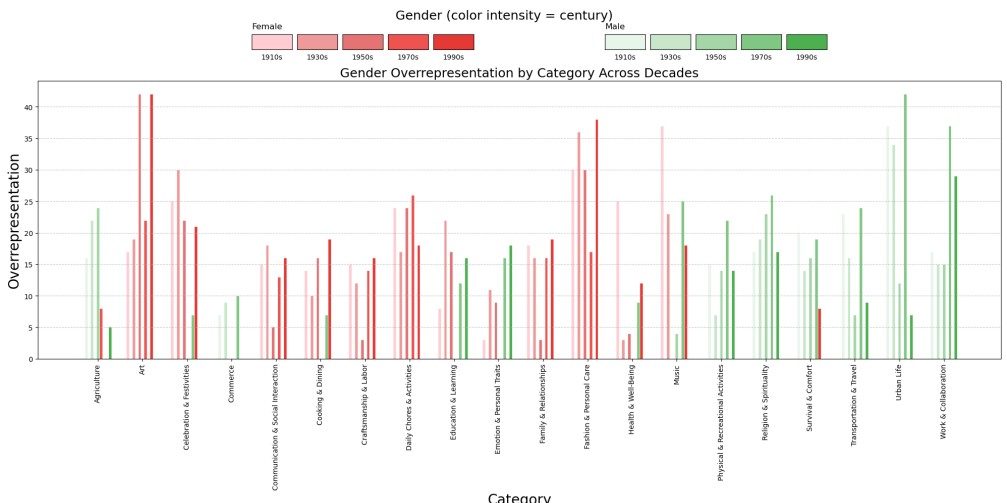

Figure 26: Over- and underrepresentation of gender predictions across **decades** for **SDXL**.

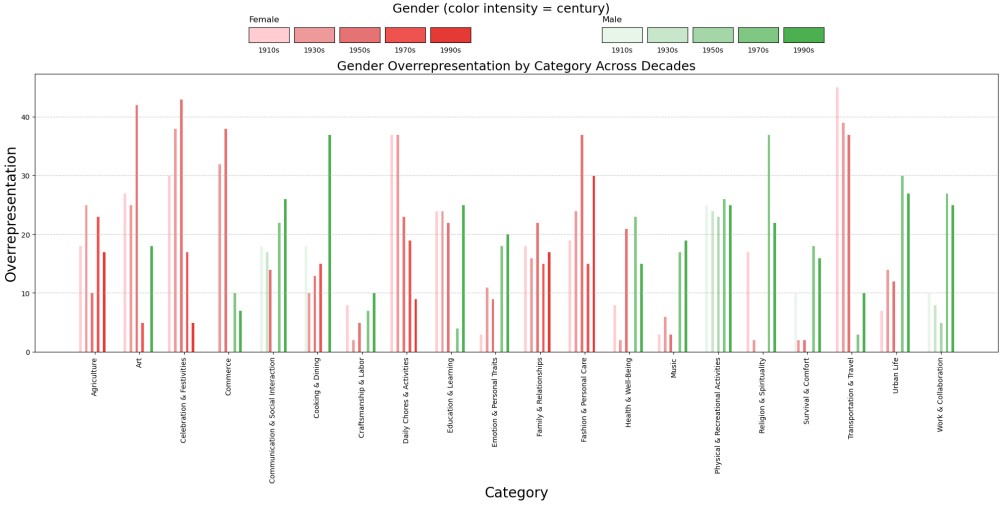

Figure 27: Over- and underrepresentation of gender predictions across **decades** for **FLUX.1**.

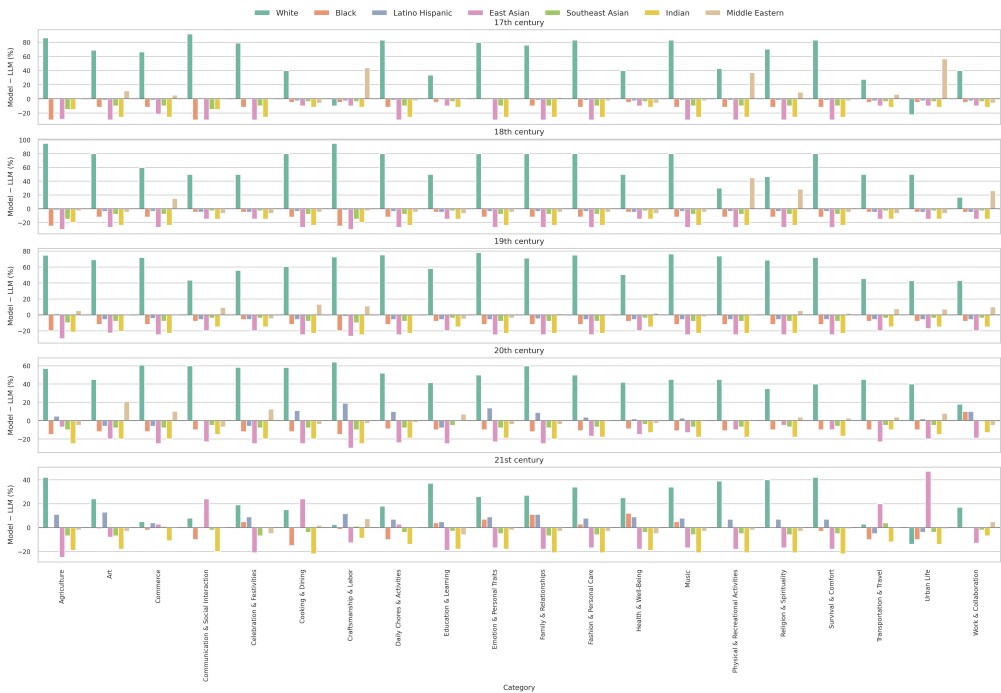

Figure 28: Racial over- and underrepresentation across centuries for **FLUX.1**.

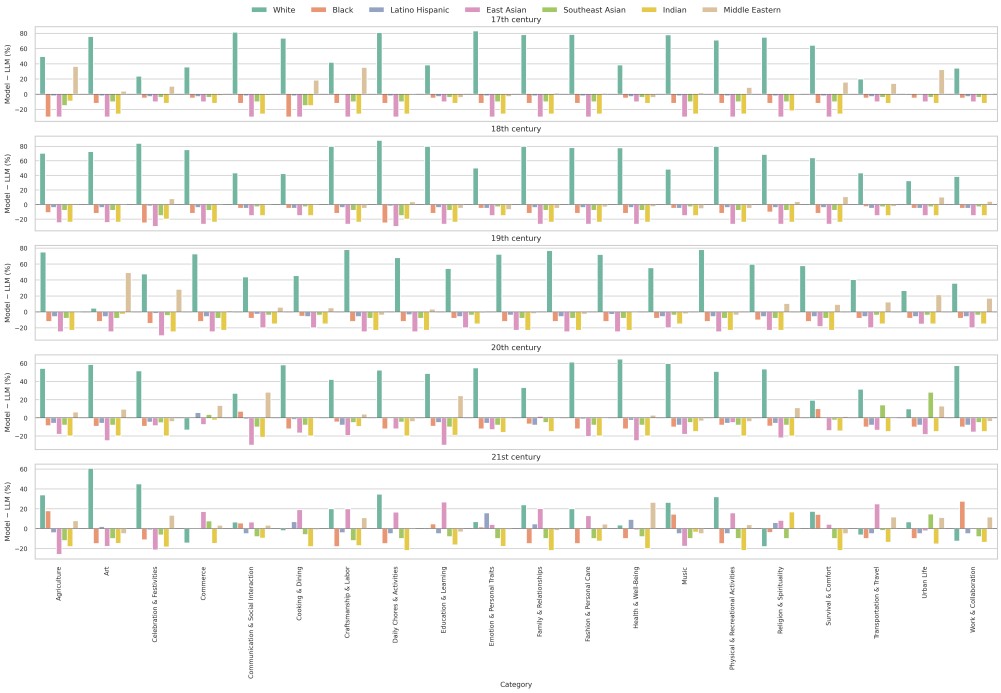

Figure 29: Racial over- and underrepresentation across centuries for **SD3**.

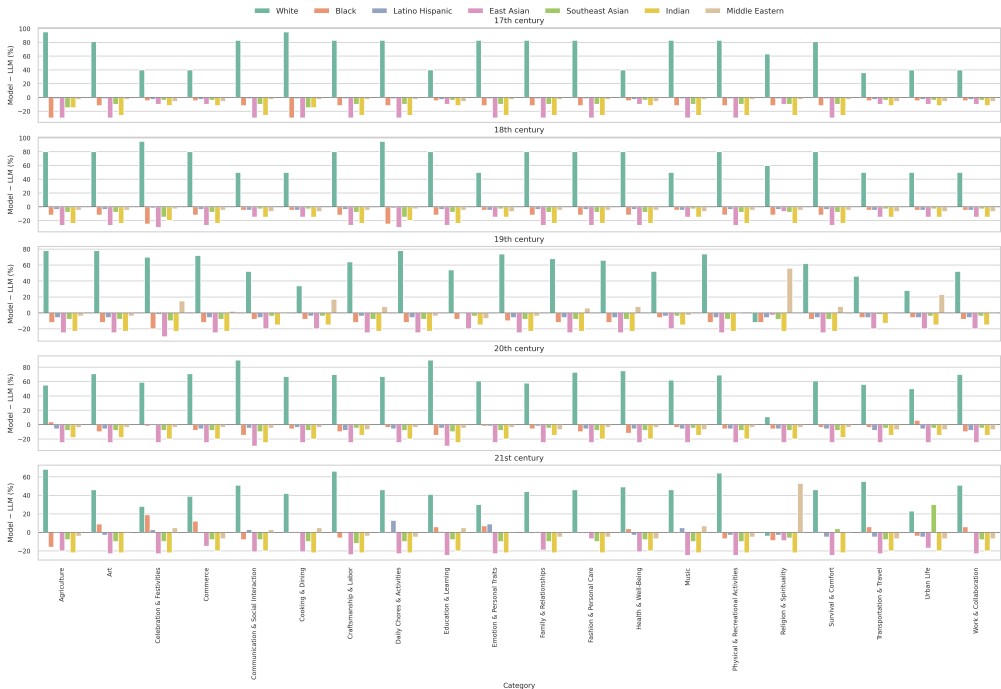

Figure 30: Racial over- and underrepresentation across centuries for **SDXL**.

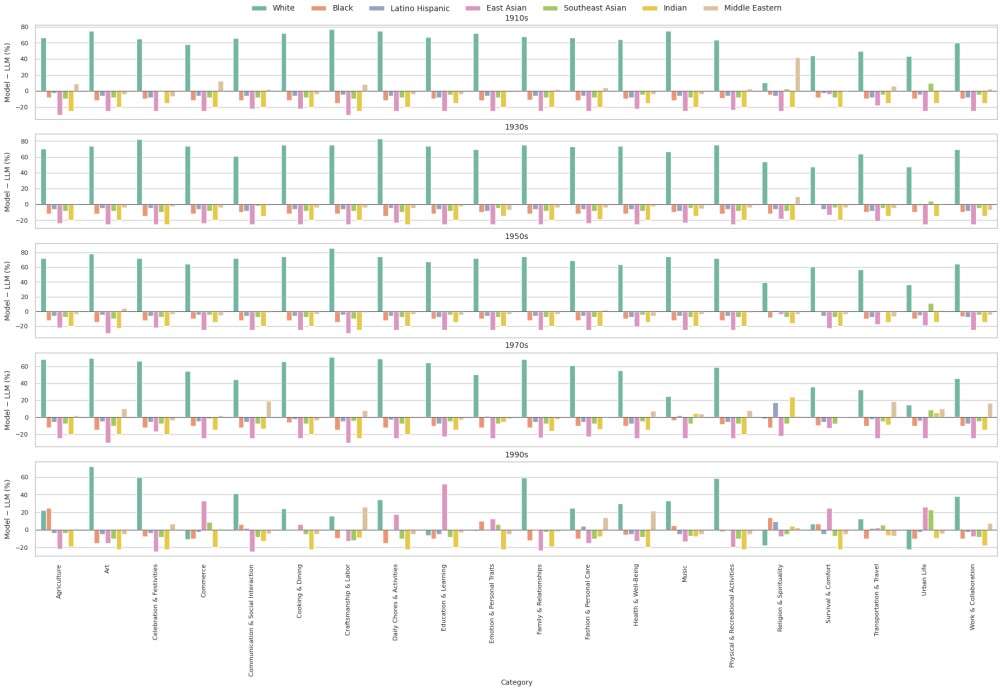

Figure 31: Racial over- and underrepresentation across decades for **SD3**.

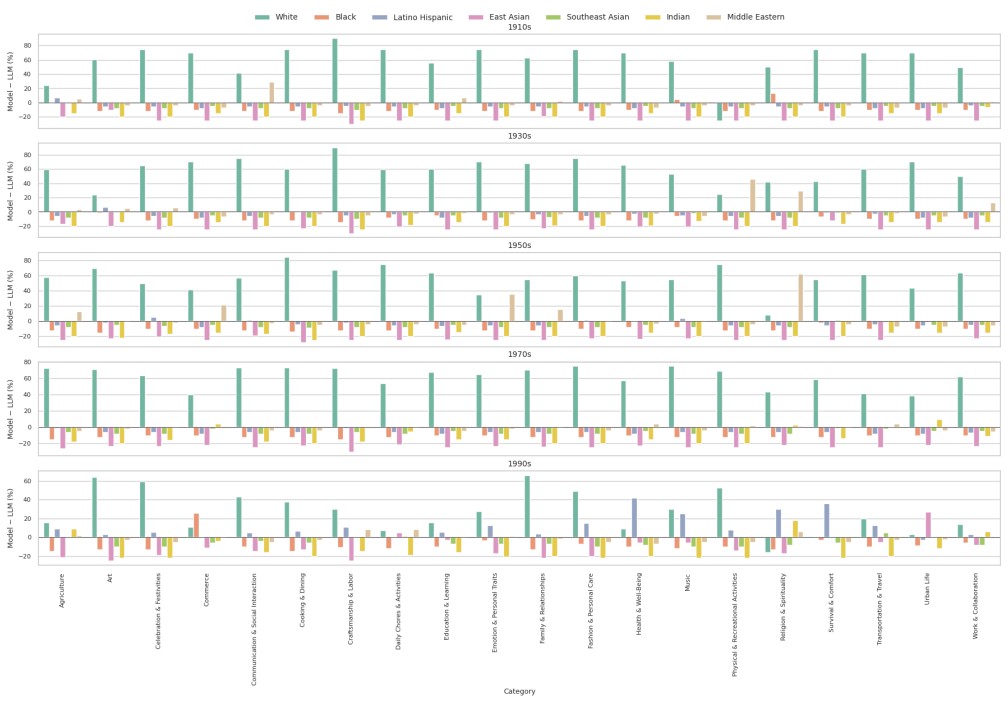

Figure 32: Racial over- and underrepresentation across decades for **FLUX.1**.

