# OpenReview forum: "Synthetic History: Evaluating Visual Representations of the Past in Diffusion Models"
_ICLR.cc/2026/Conference — ICLR 2026 Poster_

### Official Review · Reviewer_WUC2 · 2025-10-30

**Soundness:** 3
**Presentation:** 3
**Contribution:** 2
**Rating:** 6
**Confidence:** 4

**Summary:**

This paper introduces a benchmark for evaluating how text-to-image diffusion models represent historical contexts, addressing a gap in current research which has focused primarily on contemporary demographic and cultural biases. The authors created HistVis, a dataset of 30,000 synthetic images generated by three state-of-the-art models (SDXL, SD3, and FLUX.1) using neutral prompts describing universal human activities across five centuries and five decades of the 20th century. They evaluate these images along three dimensions: (1) Implicit Stylistic Associations, finding that models strongly default to specific visual styles for certain eras (e.g., engravings for the 17th-18th centuries, monochrome photography for early 20th century decades) even without explicit stylistic instructions; (2) Historical Consistency, using an automated LLM+VLM pipeline to detect anachronisms; and (3) Demographic Representation, comparing generated gender and racial distributions against LLM-derived historical baselines. The findings demonstrate that T2I models systematically struggle with historically accurate representations, relying on learned stylistic conventions and failing to properly condition on temporal context.

**Strengths:**

* First systematic evaluation of historical representation in T2I models,  articulating why this matters beyond factual accuracy—historical imagery shapes cultural memory, collective identity, and public understanding of the past, with real consequences as these systems increasingly generate educational and cultural content.
* Dataset contribution: HistVis dataset with 30,000 images across 3 state-of-the-art models (SDXL, SD3, FLUX.1), using 100 universal, temporally-agnostic activities paired with 10 time periods—this design cleverly isolates models' internal historical representations by avoiding historically-specific prompts that could encode external assumptions.
- Authors check that prompt engineering fails to mitigate biases. Mitigation experiments demonstrate that explicit instructions (adding "photorealistic" to prompts, using negative prompts to discourage monochrome) largely fail to override models' stylistic defaults
* Authors compare multiple state-of-the-art models and find systematic differences. Comparative analysis reveals model-specific failure modes (SD3 exhibits highest anachronism rates at 20-25%, SDXL most historically accurate, FLUX.1 intermediate)

**Weaknesses:**

- I think this is a good paper, but I am worried about the fact that demographic baseline uses LLM as "ground truth". The third metric relies entirely on GPT-4o to estimate historically plausible demographics, meaning any biases the LLM has will be encoded into the benchmark and treated as "correct" historical representation. This is particularly dangerous because: (1) LLMs are trained on internet data that reflects contemporary biases and incomplete historical records, not peer-reviewed historical scholarship; (2) the validation against Our World in Data only covers 3 out of 20 activity categories, leaving 85% of the benchmark unvalidated against any formal historical source; (3) even the validated categories use continent-level distributions while the actual evaluation uses race categories, introducing an additional unsupported mapping; (4) future work citing this benchmark may treat these LLM estimates as authoritative baselines, perpetuating and legitimizing whatever biases GPT-4o encoded. The authors acknowledge this is a "coarse approximation" and cannot replace expert historians, yet they publish quantitative over/under-representation scores without consulting actual historical demographers or using primary historical sources. This creates a circular validation problem where one AI system (GPT-4o) judges another (SDXL/SD3/FLUX), with no external ground truth. A benchmark critiquing historical accuracy should itself be grounded in rigorous historical methodology, not LLM outputs. The stylistic and anachronism metrics are well-validated, but the demographic analysis risks doing more harm than good by establishing flawed baselines as reference standards.

**Questions:**

- Why not use actual historical sources for demographic baselines? You validate 3 categories against Our World in Data with reasonable results (MAE=4.64 for GPT-4o). Why not extend this approach to the remaining 17 categories by consulting historical demographers, census data, labor statistics, or peer-reviewed historical scholarship? Even if comprehensive data doesn't exist for all activity-period pairs, wouldn't partial coverage with real historical data be more valuable than complete coverage with LLM estimates? What were the practical constraints (time, cost, expertise access) that prevented this?
- Did you consult any historians or historical demographers during this work? If so, what was their feedback on using LLM-generated baselines? If not, would you consider adding expert validation in a revision, at least for a subset of high-impact categories (e.g., education, work, agriculture) that future users might cite most frequently?
- How do you recommend future work should use the demographic metric? Given that you acknowledge LLM baselines are "coarse approximations" and cannot replace expert knowledge, should future papers cite the demographic over/under-representation scores as evidence of bias? Or should this metric be treated as exploratory/preliminary pending validation with real historical data? Would you consider adding stronger cautionary language in the camera-ready version?

**Details Of Ethics Concerns:**

- Tension between historical fidelity and counter-stereotypical representation. This work evaluates T2I models by measuring whether they match historical demographic patterns, including patterns that reflect past discrimination and exclusion (e.g., women concentrated in caregiving roles, men dominating technical/leadership positions, predominantly white representation in certain contexts). The paper frames deviations from these historical patterns as "biases" that models should avoid. However, this creates a fundamental ethical tension: Should T2I models faithfully replicate historical injustices, or is it beneficial—even preferable—for them to generate more diverse, counter-stereotypical imagery that depicts women in technical roles, diverse racial representation in leadership, etc.?
- The benchmark risks legitimizing historical stereotypes as "correct." By encoding LLM-estimated historical demographics as ground truth baselines and publishing quantitative "over-representation" scores when models show, for example, more women in education than historically plausible, the dataset implicitly marks historical exclusion and discrimination as the "right" way to represent the past. Future users could cite this benchmark to argue that models should be less diverse to be more "historically accurate," potentially weaponizing historical accuracy against representation goals. This is especially concerning given that the baselines come from an LLM rather than careful historical scholarship—any biases in GPT-4o's understanding of history (e.g., male-biased historical records, Western-centric perspectives) become encoded as "truth."

I am not an expert in fairness, bias, and representation ethics, so I cannot fully judge the implications

---

> ### Author Response · Authors · 2025-11-21
>
> We thank the reviewer for pointing out the weaknesses of our approach in a truly insightful manner.
>
> We understand the general concern and are aware of the issue of circular validation, but in the current setting our only choice is to keep this evaluation as an indicator of extreme deviations, as no other feasible approach exists for an initial exploration of bias in the context of our dataset of activities and periods. We were unable to find historical sources that could serve as ground truth for all categories and periods. It was equally difficult, within the scope of our work, to assemble a team of experts (since evaluating demographic distributions across all activities and periods is beyond the expertise of any single person) whose knowledge we could leverage to develop a rigorous historical methodology. Although we are aware of the problematic aspects, we have decided to include this aspect of analysis as a preliminary indicator of demographic patterns that would otherwise remain unexplored. However, we fully agree that we need to describe our approach more carefully.
>
> To clarify and take a more cautious stance, we have rewritten the introductory paragraph for this section in our revised version and have also changed throughout the text that these are not historically plausible, but “LLM-estimated historically plausible” demographics. We have also expanded our final section on limitations to include further discussions and limitations indicated by the reviewer.
>
> Below, we address each specific concern:
>
> - W1 - although LLM-derived estimates are imperfect and reflect the biases of their training data, they provide access to a form of aggregated historical knowledge that would be extremely difficult to compile manually for our defined categories of activities and periods, enabling us to derive rough and imperfect but scalable approximations that otherwise would not be feasible at all.
>
> - W2 - the OWID comparison was used to test whether LLM-derived estimates make sense in settings where some historical statistics do exist. These 3 categories were the only ones for which global demographic data could be identified. For many other activities in the benchmark, no consolidated demographic records exist at any scale. So we performed this limited evaluation of the LLM-estimated demographics, to check if the estimates can capture coarse historical signals where real data for comparison is available.
>
> - W3 - we are fully aware of this limitation. However, OWID reports only continent-level demographic distributions, so this was the only data closest to our setting that we can use to evaluate LLM estimates at least to some extent. We decided to evaluate the LLM in relation to continents to avoid more problematic continent-to-race mapping, which would require unsupported and misleading assumptions (e.g. how to represent the racial composition of the U.S. at different periods). To clarify this better, we have added an additional short explanation in the revised version.
>
> - W4 - we really appreciate pointing out this issue of potential misreadings in future work and we try to be as mindful as possible about it by explicitly stating in our revised version that these are not historical estimates, but LLM-derived historical estimates.
>
> - Q1& Q2 - We could not find census data that would provide ground truth for all activities. Informal consultation with our history department colleagues confirmed that expert-level validation across such diverse subjects would require a team of scholars specializing in both the topics and quantitative/“distant reading” approaches, which was beyond our current scope. However, we decided to keep this section, even in this unvalidated form, also with the hope to attract attention and critical engagement from historians, finding potentially future collaborators, but also inciting a discussion on LLM estimates of historical knowledge. Q3 - this metric should be treated as preliminary validation, and we have rewritten the paper to include more cautious wording.
>
> Ethical concerns - We thank the reviewer for raising this important point, which is indeed challenging not only for historically situated generated imagery but for bias mitigation in general. We have updated our revised version to include a separate section (7.2) dedicated to this issue that: 1) highlights the complexity of balancing accurate historical depiction with avoiding the propagation of biases and exclusionary narratives, and 2) emphasizing that our LLM-derived baselines do not represent an expected standard. While in our analysis we use the terms “over” and “under” representations, we included this section to highlight that our baseline does not necessarily represent a required type of behavior. We specifically indicate that “Our intention is not to prescribe a “correct” way of depicting the past, but to provide a starting point for addressing this challenge at a broader, systematic scale.”

---

### Official Review · Reviewer_2DRT · 2025-10-31

**Soundness:** 3
**Presentation:** 2
**Contribution:** 3
**Rating:** 6
**Confidence:** 4

**Summary:**

The paper presents a framework for evaluating text-to-image models in terms of their ability to accurately represent historical context. The paper presents HistVis, a dataset of 30,000 images generated from three open-source diffusion models that were prompted to depict people performing generic activities across different centuries and decades. They then propose an evaluation protocol that examines stylistic associations, historical consistency, and demographic distribution in the generated images, comparing the results with historical data. The study reveals interesting patterns in how models capture or distort aspects of history, offering insights into biases related to style, anachronism, and representation. Overall, this work highlights the lack of historical accuracy in generative models and provides a concrete methodology and dataset to study it.

**Strengths:**

- The paper highlights an important problem of evaluating historical representation in text-to-image models when depicting generic, everyday activities and provides a clear motivation for addressing it.
- The paper provides and evaluates an interesting dataset consisting of images that depict a comprehensive set of timeless activities spanning approximately five and a half centuries, offering strong coverage across diverse historical periods.
- The findings, particularly the observation of anachronistic objects in images depicting historical time periods, are very interesting and effectively highlight an important issue in the temporal consistency of these models.

**Weaknesses:**

- Although the VSG score is supported by a robust methodology, the reason for evaluating biases in style associations and the explanation of the distinct style classes are not sufficiently motivated.
- The anachronism detection uses an LLM to get a list of possible objects that could be anachronistic in a given activity. How can we ensure that this list is exhaustive? Were other methods, like object detection, considered for this task?
- Evaluating demographic representation in the generated stories is an interesting direction. However, the use of LLMs to predict demographic distributions raises concerns about reliability. The paper uses the public OWID dataset to demonstrate the robustness of GPT-4o as an estimator for a small set of activities but does not account for potential data contamination, which could explain the model’s high accuracy in these limited cases.
- Figures 4 and 6 could be significantly improved to enhance clarity and effectively communicate the key takeaways. In their current form, they present a large amount of information in a single view, which makes it difficult for readers to interpret the results and understand the main insights.

With the style association results, it might be interesting to link them to the training data and see correlations/reasons for such biases. (Although it might be out of scope for this paper)

Minor typos (not really a weakness, sharing so that authors can fix them):
- Line 275 refers to Appendix 5 instead of Table 5
- Line 346 uses the word “currentumptions”.
- Line 439 says Section Q, instead of Appendix Q

**Questions:**

- Is the stylistic predictor (to predict the style class) also trained to guess images that are not any of the 5 categories in the training dataset? As a mitigation prompt was used, what was its goal?
- The paper notes that clothing appears in 2–5% of the anachronistic images (line 322). It is unclear why clothing is considered anachronistic. Does this refer to attire that does not match the depicted time period? If so, it would be helpful to clarify how such inconsistencies were detected.
- Was the prompt used to obtain the racial breakdown for a particular time period specified by continent (as the OWID comparison was based on continent)? If so, which continent were the proportions of the generated images compared against? If not, was it assumed that GPT-4o would estimate global demographic proportions for the corresponding historical period?

---

> ### Author Response · Authors · 2025-11-21
>
> We thank the reviewer for their insightful comments.
>
> Weaknesses:
>
> - We thank the reviewer for pointing this out and have revised our paper to clarify the motivation. We analyze stylistic biases because they meaningfully shape how models visually construct historical periods and can reinforce  stereotyped assumptions about what the past “should” look like. As shown in our mitigation experiment, these stylistic defaults are often not overridden even when explicitly requesting different styles. This motivation was originally discussed in Appendix C and is now incorporated directly into the updated main text (lines 166–168) for clarity.  The five style categories were selected because they consistently emerged during qualitative inspection of both reference and generated images. While not exhaustive, these categories capture the main stylistic trends relevant to historical depiction. We agree that future work could refine or expand this taxonomy, and we have added this point to the limitations and highlighted it as a future direction in the revised Discussion.
>
> - We agree that the LLM-generated list of potentially anachronistic objects cannot be fully exhaustive and have updated the paper to include this observation. We tried to capture the most frequent and clearly identifiable anachronisms rather than every possible historical inconsistency. In our revised version we explicitly acknowledge the fact that not all possible anachronisms are included as in the section 7.1.We also considered object-detection–based approaches, but these rely on fixed, modern taxonomies and would substantially limit the generality of the method.
>
> - We are aware of the concerns related to the use of LLM to predict demographic distributions. We have updated our paper to provide a more cautious framing regarding our setting as well as to include a discussion about the various limitations.  We address the issues related to the use of LLM for demographic estimates in detail in our response to Reviewer WUC2. As for the issue of data contamination, we do not consider it necessary to be a problem in this context.  In well-documented areas, some overlap with the model’s training data is expected, and accurate matches suggest that the LLM is relying on real historical patterns rather than hallucinating. This validation step helps confirm that the model aligns with known historical trends where reliable data is available.
>
> - We agree that Figures 4 and 6 contain a lot of information, as each visualization integrates results spanning multiple periods, activities and demographic categories. But we wanted to condense these complex patterns into a unified view, even if it is dense. In the revised version of our paper we enlarged both figures and improved their readability to make the key trends easier to follow, while preserving the single-view format that highlights how these dimensions interact.
>
> Training Data Correlations: We agree that linking style defaults to specifics of the training data would be an interesting research direction, but the training data for the models we evaluate is not publicly available, making such an analysis currently infeasible.
>
> Questions
>
> - (a) No, the predictor is trained only on the five style categories described in the manuscript. As for the mitigation prompt, its purpose was to test whether the dominant stylistic defaults could be overridden through prompt engineering. If simple instructions (e.g. “photorealistic”) would be enough to change the style, these biases would be less problematic. But we found that models frequently could not override these default styles for earlier periods, even under explicit guidance.
>
> - Yes, “clothing” refers specifically to attire that does not match the depicted historical period, e.g. T-shirts, modern athletic wear, or contemporary footwear appearing in images meant to represent pre-modern centuries. An illustrative example is provided in Appendix I (“A person exercising in the 18th century”), where the detection of modern trainers was categorized under the broader umbrella of modern clothing.
>
> - No, the prompt used to obtain the racial breakdown for a particular time period was not specified by continent. We designed a different set of prompts only for the evaluation of LLMs, and this set includes only continent-level breakdown, while our main set of prompts includes only race. Because OWID does not provide race-based statistics, only continent-level and this was the only existing data closest to our setting that we can use to evaluate LLM estimates at least to some extent. To avoid oversimplified and problematic continent–race mappings, the evaluation of LLMs replaces race estimation with continent-level estimation. . We have updated our revised version of the paper to make this clearer (Section 6.1.) and we added the exact evaluation prompts in Appendix O.

---

### Official Review · Reviewer_7quq · 2025-11-01

**Soundness:** 4
**Presentation:** 4
**Contribution:** 3
**Rating:** 8
**Confidence:** 3

**Summary:**

This paper conducts a study over biases of text-to-image diffusion models while generating images of historical ages. The evaluation criteria is three fold: 1) examining stylistic bias, 2) historical consistency, 3) demographic representation. The authors conducted in-depth evaluation and analysis over each of these aspects. They contributed the HistVis dataset that has 30K synthetic images generated by 3 T2I models -- SDXL, SD3, Flux.1Schnell.

**Strengths:**

The paper is a great read, easy to follow, with interesting findings and extensive evaluations. The authors have designed careful and sound evaluation schemes for each of the three aspects that they are studying in the paper. I especially liked that they have used multiple VLMs for evaluation rather than only using one as a judge. All details of the study has been laid out in complete transparent detail. Multiple qualitative examples were very helpful in getting the point across for each aspect of the study. The authors also discuss and analyze their findings in detail which gives readers valuable insights.

**Weaknesses:**

A comparison with related studies in this direction comparing the number of samples and evaluation strategies will be helpful to better place the paper.

**Questions:**

Are the HistVis dataset prompts all manually designed or were LLMs involved in aiding ideation or prompt design? It will be interesting to know how the authors designed these prompts.

---

> ### Author Response · Authors · 2025-11-21
>
> We thank the reviewer for their positive review and appreciation of our contribution.
>
> Weakness:
>
> - To the best of our knowledge, our work is the first to propose a dedicated benchmark for evaluating historical representation in TTI-generated imagery. In the Related Work section, we have incorporated all major lines of research relevant to this area, including studies on demographic, cultural, and geographic biases in diffusion models; early investigations of historical sensitivity in multimodal captioning systems; datasets evaluating the recognition of historical figures and landmarks; and perspectives from media studies on AI-generated historical aesthetics. However, no prior work has systematically examined how generative models contextualize and visually encode historical periods across diverse activities. Our benchmark is intended to fill this gap. That said, if there are additional studies the reviewer believes would further strengthen this section, we would be very grateful for the pointers and would be happy to integrate them into the final version.
>
> Question:
> - We thank the reviewer for this question. All HistVis prompts follow a predefined template (“a person [activity] in the [historical period]”), and the set of historical periods was specified by the authors in advance. LLMs were used at the ideation stage to suggest possible activity candidates; the authors then manually curated and finalized the activity list. Appendix B (HistVis Prompt Design) has been updated to explicitly clarify this process.

---

### Official Review · Reviewer_Eaph · 2025-11-01

**Soundness:** 3
**Presentation:** 3
**Contribution:** 3
**Rating:** 4
**Confidence:** 4

**Summary:**

This paper evaluates the historical knowledge embedded in text-to-image (T2I) models. The authors generate images using multiple T2I models based on predefined actions and time periods, ranging from the 17th century to the late 20th century. The analysis focuses on three key aspects: implicit stylistic associations, anachronism detection, and demographic representation. Using carefully defined metrics for each aspect, the study reveals several noteworthy findings: a) different TTI models exhibit distinct stylistic associations across time periods, and these associations persist even under explicit prompting; b) the models frequently produce anachronistic elements, reflecting a lack of temporal awareness; and c) the generations display gender and racial over- or under-representation, highlighting underlying demographic biases. While the intent behind each such direction is appreciated, i have some issues with the methods used to evaluate these, which i have summarized in the weaknesses.

**Strengths:**

1. The paper explores the world knowledge embedded in text-to-image (TTI) models—knowledge that parallels that of modern Large Language Models (LLMs). By examining how these models represent historical contexts, the authors go beyond the typical focus on creativity or imagination to probe their practical understanding of reality. This represents an important and relatively underexplored research direction.
2. The findings reveal previously unexamined layers of bias within TTI models. For instance, the recurring depiction of modern artifacts such as headphones in images set in the 17th or 18th century underscores the models’ limited grasp of historical realism and their tendency to fill knowledge gaps with contemporary concepts. Addressing such issues is crucial for improving the reliability and historical awareness of future model releases.
3. The paper is clearly structured and well written, making complex ideas accessible and easy to follow.

**Weaknesses:**

1. The paper’s positioning could be clearer. The provided dataset, being composed of the outputs from T2I models applied to a set of prompts, offers limited standalone value to the community—apart from ensuring reproducibility. The true contribution appears to lie in the methodological framework for analyzing the historical biases in generative models. It would therefore strengthen the paper to explicitly present the work as proposing a benchmark for estimating VLM biases in representing historical contexts, with the accompanying dataset serving as an illustrative application of this benchmark to three specific models.
2. In the anachronism detection evaluation, the reported 72% alignment with human judgment is substantially low, casting some doubt on the robustness of the anachronism detection component. The concern here is less about the existence of anachronisms and more about the quantitative reliability of the reported metrics.
3. The use of Large Language Models (LLMs) to measure gender and racial representation raises validity concerns. While the paper attempts to verify these measures through domains such as education and agriculture, the scope of the evaluation appears insufficient to justify LLMs as reliable tools for assessing historical demographic biases. A more cautious approach would be to limit this analysis to tasks with concrete supporting data and to avoid the use of LLM-based estimations where empirical grounding is weak.

**Questions:**

The paper uncovers stylistic biases across time periods, however, these would be expected on some levels due to the data available on those time periods (portraits and illustrations in older days vs black and white in 1900s vs more varieties nowadays). May be the issue can be avoided just by prompting the models to generate only real-life images. What's the authors' take on this?

---

> ### Author Response · Authors · 2025-11-21
> **Response to Reviewer Eaph**
>
> We appreciate the reviewer's thoughtful feedback on our manuscript. Below are our answers to the specific weaknesses raised:
>
> Weaknesses:
>
> - Our intention is indeed to present this work as introducing a benchmark for evaluating historical representation in TTI models, with HistVis functioning as the dataset component that supports this benchmark. We believe this framing is already present throughout the paper: the abstract states that “we introduce a benchmark … The benchmark combines HistVis … with a reproducible evaluation protocol” (Abstract, lines 15–19); Section 1 explains that “we construct a benchmark that pairs HistVis with a reproducible evaluation protocol” and stresses that the framework is model-agnostic and applicable to any TTI model (Section 1, lines 79–85); and Section 3 reiterates that “as part of our benchmark, we introduce HistVis …” (Section 3, lines 144–145). If there are specific passages where our positioning appears unclear or could be strengthened, we would be grateful for the guidance and will revise the final version accordingly.
>
>
> - We would appreciate clarification on the reference to the 72% agreement figure, as that value reflects only GPT-4o results. In our evaluation, we rely on majority voting across three VLMs (GPT-4o, LLaMA-3.2, and Qwen2.5), which increases alignment with human judgments to 75%, while remaining fully reproducible with open-source alternatives. We also note that historical anachronism detection is inherently interpretive, particularly for ambiguous cases such as clothing styles or backgrounds without clear temporal markers, and even human annotators do not fully agree (Fleiss’ κ = 0.63, indicating substantial but not perfect agreement). Based on the reviewer's comment and to provide more context, in our revised version we have added an additional reference to prior work such as OpenBias [1], which reports 67% agreement on simpler, contemporary VQA tasks (e.g., identifying an object’s color). Viewed in this light, we believe that 75% agreement in a semantically more complex setting represents a strong result.. We thank the reviewer for encouraging us to expand on this point with more context.
>
>
> - We appreciate this concern and fully agree that LLMs are not substitutes for historical data. To clarify this in the manuscript, we revised the introductory paragraph of the demographic section, adopted the term “LLM-estimated historically plausible,” and expanded the limitations section to explicitly acknowledge the risks noted by the reviewer. Our intention is not to treat them as historical authorities, but to use them as coarse, scalable heuristics for detecting extreme demographic deviations in TTI outputs. The metric is diagnostic rather than prescriptive: it does not aim to define what is “historically correct,” but flags cases where generated demographics diverge so strongly from any reasonable prior that they likely reflect model behavior rather than meaningful engagement with historical context. Restricting the analysis only to domains with concrete demographic records would narrow the benchmark’s prompt-agnostic scope and prevent evaluation across the many activities and periods for which no demographic baseline exists. Accordingly, the OWID comparison serves only as a plausibility check where limited data are available, not as validation of LLM-derived estimates as historical ground truth.
>
>
> Questions:
>
> - We agree that stylistic differences across historical periods are to be expected, given the types of images that survive from those eras, and these differences are not inaccuracies on their own. Our concern is that these styles seem to operate as persistent generative defaults that users often cannot override. Similar patterns have been noted in prior work, where models default to stylized or cartoon-like depictions of non-Western artifacts [2]. Specifically, our experiments with SDXL showed that even when given explicit instructions for photorealism and negative prompts against monochrome output, the model largely retained its default historical styles (e.g., engraving tendencies for earlier centuries). Appendix E includes representative examples of this experiment. This suggests that the model's preferred style is a hard-coded bias, not just a neutral default, which severely restricts users' ability to generate alternative or counterfactual historical visualizations. To make this motivation clearer, we have now moved part of the explanation previously in Appendix C into the main text (lines 166–168).
>
> [1] Moreno D’Incà et al. OpenBias: Open-set bias detection in text-to-image generative models. In Proceedings of CVPR 2024.
>
> [2] Nithish Kannen Senthilkumar et al. Beyond aesthetics: Cultural competence in text-to-image models. NeurIPS, 37, 2024.

---

> > ### Comment · Reviewer_Eaph · 2025-11-24
> >
> > I appreciate the responses to my concerns. Most of my concerns have been addressed.
> >
> > **The metric is diagnostic rather than prescriptive: it does not aim to define what is “historically correct,” but flags cases where generated demographics diverge so strongly from any reasonable prior that they likely reflect model behavior rather than meaningful engagement with historical context.**: I agree this is not to measure correctness, but about comparison with some approximate prior. In the absence of ground truth, probably comparing with the uniform distribution of genders/races would have been more prudent, and then analyzing the gender/race diversity across time periods to see if they change over time (e.g., scientists in the 18th century would have 0 females, 19th century may have 5% females, 20th century may have 10-20% females etc). Moreover, gender and racial biases are rampant wrt normal prompts as well. Your baseline distribution could have been that of images generated from normal prompts like 'photo of a scientist', which would have told you how much bias is being added due to an earlier time period.
> >
> > **Anachronism related human alignment**: Would this improve if the authors were provided the information on historical facts (like headphones were invented only in year ****)? I understand this may be infeasible at scale, but were participants allowed to surf the internet before annotating? that might have helped increase the fleiss kappa.

---

> > > ### Author Response · Authors · 2025-11-26
> > >
> > > Demographic Evaluation: We thank the reviewer for their thoughtful recommendations, and we are glad to hear that most of their concerns have been addressed. We agree with the reviewer that a uniform distribution might seem as a more prudent initial setup, but then “the burden of proof” shifts to the level of interpretation and we are again confronted with the same issue of how to evaluate the resulting over- and under-representations in relation to a uniform distribution, especially if we are interested not only in how they change over time (e.g. 18th century  0 females, 19th century 5% females, 20th  10-20% females), but how plausibly they represent a specific time period. Similarly, the suggestion of using a neutral prompt without historical conditioning as a baseline for the bias distribution (e.g. just “the photo of a scientist”), would indicate whether the historical conditioning changes the baseline bias, but not if the changes are meaningful and plausible for the given time period.
> > >
> > > This is why we decided to use LLM-estimated distributions as our baseline, not as a historical ground-truth, but to encode potentially plausible directional expectations, for example, to indicate that women scientists were nearly absent in the 18th century but increasingly present by the 20th. While these estimates are necessarily coarse, they still seem to provide a useful preliminary estimation because our comparison with OWID suggests that, at least where records exist, they approximate known patterns reasonably well for this diagnostic purpose.
> > >
> > > Anachronism related human alignment: We agree that access to historical information could help annotators, and we note that participants were not given any time limit for answering. They were free to search for historical facts online if they wished, so they could verify details such as the invention dates of specific objects before responding.

---

### Meta-Review · Area_Chair_aAXz · 2025-12-09

**Summary:**

This paper was reviewed by 4 knowledgeable referees. The reviewers found the research direction covered in the paper important and under-explored (Eaph, 2DRT, WUC2), with a clear motivation and insightful findings (Eaph, 7quq, 2DRT) and extensive evaluations (7quq, WUC2). The reviewers raised concerns about:
1. The clarity of the contribution: emphasis on the dataset vs. the benchmark and analysis (Eaph)
2. The alignment with human judgement of anachronism detection (Eaph)
3. The use of LLMs to predict demographic distributions (Eaph, 2DRT, WUC2) and to obtain of anachronistic objects in a given activity (2DRT), and as a result the strength of the claims made in the paper (WUC2)
4. The positioning of the paper w.r.t. related studies (7quq)
5. The motivation for evaluating biases in style associations was not clear (2DRT)

**Reviewer Concerns:**

The rebuttal argues the positioning of the contribution as a benchmarking and analysis yielding interesting observations, clarifies the motivation about stylistic biases, and suggests this is the first work to propose a dedicated benchmark for evaluating historical representation in TTI. The authors' response also addresses the computation of alignment with human judgement by explaining the followed majority voting procedure and by contrasting with results reported in prior work on simpler tasks. The rebuttal acknowledges the limitation of leveraging LLMs to predict demographic distributions and argues that finding historical sources as ground truth for all categories and period is challenging and therefore validation was done on those where such data was available. To address this concern, the authors introduce an extended limitations paragraph to the paper and adopt a more cautious framing.


Although the rebuttal mentions positioning the work w.r.t. all other bias dimensions studied in the literature, the AC would like to point out the missing references which should be considered for the final version of the manuscript [a,b,c,d].

[a] https://aclanthology.org/2025.findings-emnlp.1141/
[b] https://dl.acm.org/doi/abs/10.1145/3630106.3658927
[c] https://arxiv.org/abs/2308.06198
[d] https://dl.acm.org/doi/abs/10.1145/3630106.3658967

**Reviewer Scores:**

Some reviewers participated in the discussion and acknowledged their concerns had been addressed. Other reviewers didn't have the opportunity to follow up with the authors. The recurrent concern raised by the reviewers was related to the use of LLMs and the rebuttal adequately addressed this concern, hence, the AC thinks reviewers might have been willing to increase their scores.

---

### Decision · Program_Chairs · 2026-01-26

Accept (Poster)